:ᴏ: PLOS ONE

# Manual dexterity of mice during food-handling involves the thumb and a set of fast basic movements

**John M. Barrett** [iD]*, **Martinna G. Raineri Tapies, Gordon M. G. Shepherd**

Department of Physiology, Feinberg School of Medicine, Northwestern University, Chicago, Illinois, United States of America

* john.barrett@cantab.net

## Abstract

The small first digit (D1) of the mouse's hand resembles a volar pad, but its thumb-like anatomy suggests ethological importance for manipulating small objects. To explore this possibility, we recorded high-speed close-up video of mice eating seeds and other food items. Analyses of ethograms and automated tracking with DeepLabCut revealed multiple distinct microstructural features of food-handling. First, we found that mice indeed made extensive use of D1 for dexterous manipulations. In particular, mice used D1 to hold food with either of two grip types: a pincer-type grasp, or a "thumb-hold" grip, pressing with D1 from the side. Thumb-holding was preferentially used for handling smaller items, with the smallest items held between the two D1s alone. Second, we observed that mice cycled rapidly between two postural modes while feeding, with the hands positioned either at the mouth (oromanual phase) or resting below (holding phase). Third, we identified two highly stereotyped D1-related movements during feeding, including an extraordinarily fast (~20 ms) "regrip" maneuver, and a fast (~100 ms) "sniff" maneuver. Lastly, in addition to these characteristic simpler movements and postures, we also observed highly complex movements, including rapid D1-assisted rotations of food items and dexterous simultaneous double-gripping of two food fragments. Manipulation behaviors were generally conserved for different food types, and for head-fixed mice. Wild squirrels displayed a similar repertoire of D1-related movements. Our results define, for the mouse, a set of kinematic building-blocks of manual dexterity, and reveal an outsized role for D1 in these actions.

☞ OPEN ACCESS

**Data Availability Statement:** The tracking data and manually generated ethograms are available on GitHub at https://sourceforge.net/p/seed-handling-paper/code, along with the Python and Matlab code used to analyze the data presented in this paper.

## Introduction

Functional diversity of the forelimb is a hallmark of mammalian evolution [1]. Mammalian hands evolved around a basic skeletal structure, and the first digit (D1, pollex) often features prominently in these specializations [2]. The opposable D1 of humans and certain other primates enables small items such as food morsels to be held deftly and gently in a delicate precision grip [3]. Many neurons in motor cortex fire briskly and selectively during such

The videos and trained DeepLabCut models used to track digit locations are available at on Zenodo at https://zenodo.org/record/3531964.

**Funding:** This work was funded by grant NS061963 from the National Institute of Neurological Disorders and Stroke at the National Institutes of Health. The funders had no role in study design, data collection and analysis, decision to publish, or preparation of the manuscript.

**Competing interests:** The authors have declared that no competing interests exist.

D1-related grips [4, 5], and in somatosensory cortex the cortical representation of D1 is disproportionately large, including in rats [6, 7].

Mice are widely used to study forelimb function and dysfunction [8], but the possibility that D1 is used to handle small objects has received little experimental attention. Rather, the mouse's tiny D1, a mere ~0.5 mm in length, appears rudimentary [9]. However, rats possess a D1 that is anatomically small but musculoskeletally fully formed and equipped with a flat (rather than claw-like) nail, which may be used for pincer-like grasping during feeding [10, 11]. Squirrels exhibit an extreme form of D1 use, involving bimanual "thumb-holding" of food items between their two D1 digits [12]. In mouse species such as *Mus musculus*, D1 appears miniscule and more pad-like than digit-like, though with fully formed musculoskeletal anatomy [13, 14]. Mouse hand movements have been assessed using video-based motion tracking, but studies have focused mostly on the kinematics of the forelimb and D2-D5, not D1, and mostly for reach-and-grasp movements, not dexterous handling of small objects [15–17].

Exactly how the mouse's D1 is involved in the handling of small objects such as food items is an open question. Addressing this has potential not only to inform future motor neuroscience studies of manual dexterity in the mouse, but also to shed light on the evolution of dexterous forelimb movements in mammals more generally, and to augment the growing repertoire of tasks for studying forelimb dysfunction in animal models of motor diseases. To study this behavior, we performed video analysis of hand movements in adult mice feeding on seeds. Because we were interested in understanding naturalistic feeding-related behaviors from an ethological perspective [18, 19], in the main set of experiments we focused on observing food-handling by mice that were unrestrained, moving and feeding freely within a small chamber. Because head-fixation is an important methodology that enables neural recording and stimulation to elucidate mechanisms of forelimb motor control in mice (e.g. [17, 20, 21]), in a subset of experiments we extended these studies to head-fixed mice. Additionally, because the findings suggested similarities as well as differences with the previously reported thumb-holding behavior of squirrels [12], in a limited set of field studies we analyzed videos of wild squirrels handling food. Overall, our results reveal extensive use of the thumb and provide new insights into fundamental aspects of food-handling behavior by mice.

## Results

### Mice use their thumbs to handle food

The C57BL/6 mouse possesses a pentadactyl manus with a small D1, longer D2-D5 digits, and multiple digital and volar pads, as observed by micro-CT and macroscopy (**Fig 1**). The D1 digital and thenar pads are close together, separated by a crease-like cleft, and D1 has a flat nail (**Fig 1C, inset**).

To investigate how mice manipulate small objects, we recorded high-speed close-up videos for a cohort of mice (*n* = 8, **Table 1**) that were given seeds. In the following sections and in the main figures, we mainly present the results obtained with flaxseeds (except where indicated), with reference to additional results for wheat berries, which were broadly very similar (**S1 Fig**). A typical example of seed-handling is shown in **S1 Video**. We observed that, surprisingly, mice held seeds using a "thumb-holding" grip, in which the medial side of D1 appeared to be pressed against the side of the item (**Fig 2A and 2B**; **S2A Video**). Mice held particularly small items using only their thumbs, wedging the item into the cleft between the D1 and thenar pad (**Fig 2A**; **S2B Video**). We also observed use of a pincer-type grasp, in which D1 (and/or the thenar pad) appeared to be used together with D2 (of the same hand), with D1 positioned more posteriorly with respect to the food item (**Fig 2A and 2B**). We analyzed D1 usage and classified the grip type of each D1 as thumb-holding, pincer-type grasp, or indeterminate (**Fig**

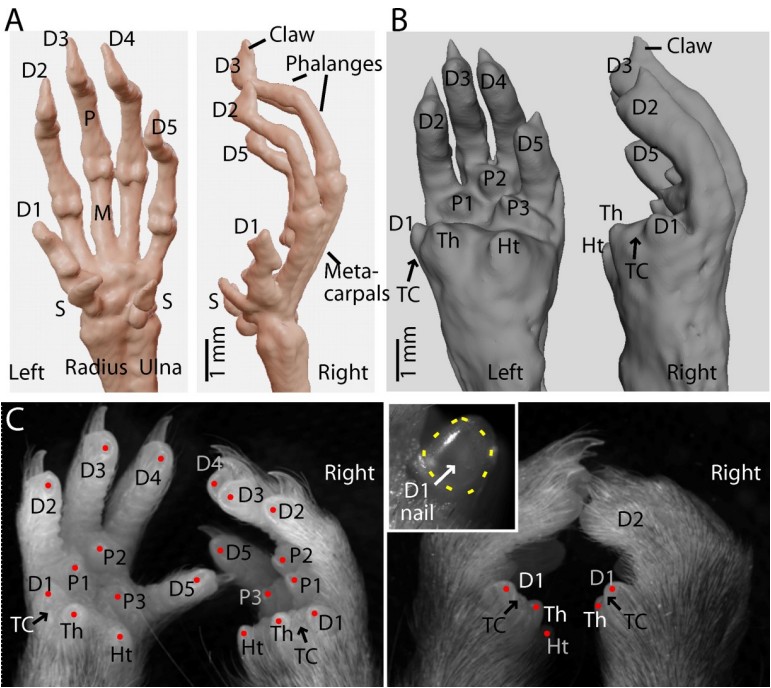

**Fig 1. Anatomy of mouse D1.** (A) Micro-CT imaging of a pair of mouse hands showing the skeletal anatomy, with labeling of digits (D1-5) and sesamoid (S), metacarpal (M), and phalangeal (P) bones. (B) Soft-tissue rendering, showing the volar aspect, with labeling of the thenar (Th), hypothenar (Ht), and interdigital pads (P1-3), and the thumb cleft (TC) between the D1 and thenar pads. (C) Left: Macroscopic images of a pair of amputated mouse hands, showing major features of the volar aspect. Right: Same, but as a dorsal view with the hands positioned more naturally. Inset: Close-up view of the flat thumb nail on the left D1.

**2C**). For flaxseeds, mice used thumb-holding more than pincer-type grasping (thumb-holding: 54 ± 8%, median ± median absolute deviation (m.a.d.), $n$ = 8 mice; grasping: 19 ± 10%; Wilcoxon's signed-rank: $W$ = 0, $p$ = 0.008**; **Fig 2D**), and grip type was usually the same for each D1, although occasionally mixed (same: 56 ± 11%, mixed: 9 ± 7%, $W$ = 1, $p$ = 0.016**, **Fig 2E**). Similar results were found for wheat berries, except that mice thumb-held less (wheat thumb-

**Table 1. Summary of the main dataset.**

| Mouse | Flaxseed | | | | Wheat berry | | | |
|---|---|---|---|---|---|---|---|---|
| | # Video segments | Total time (min:sec) | Median time | M.A.D. time | # Video segments | Total time | Median time | M.A.D. time |
| 1 | 4 | 0:45 | 0:14 | 0:02 | 1 | 0:44 | 0:44 | 0:00 |
| 2 | 2 | 0:31 | 0:16 | 0:11 | 3 | 1:37 | 0:04 | 0:00 |
| 3 | 4 | 1:19 | 0:19 | 0:10 | 2 | 1:48 | 0:54 | 0:18 |
| 4 | 2 | 0:23 | 0:12 | 0:06 | 1 | 0:25 | 0:25 | 0:00 |
| 5 | 2 | 0:26 | 0:13 | 0:01 | 1 | 0:37 | 0:37 | 0:00 |
| 6 | 8 | 0:23 | 0:03 | 0:01 | 2 | 1:05 | 0:33 | 0:21 |
| 7 | 1 | 0:41 | 0:41 | 0:00 | 1 | 1:24 | 1:24 | 0:00 |
| 8 | 2 | 0:32 | 0:16 | 0:09 | 3 | 1:07 | 0:09 | 0:08 |
| Total | 25 | 5:00 | | | 14 | 8:46 | | |

For each mouse in the main cohort (**Figs 2–5, S1–S9 Videos**) and each seed type, the number of video segments used for analysis is listed, along with the total and median length of the videos.

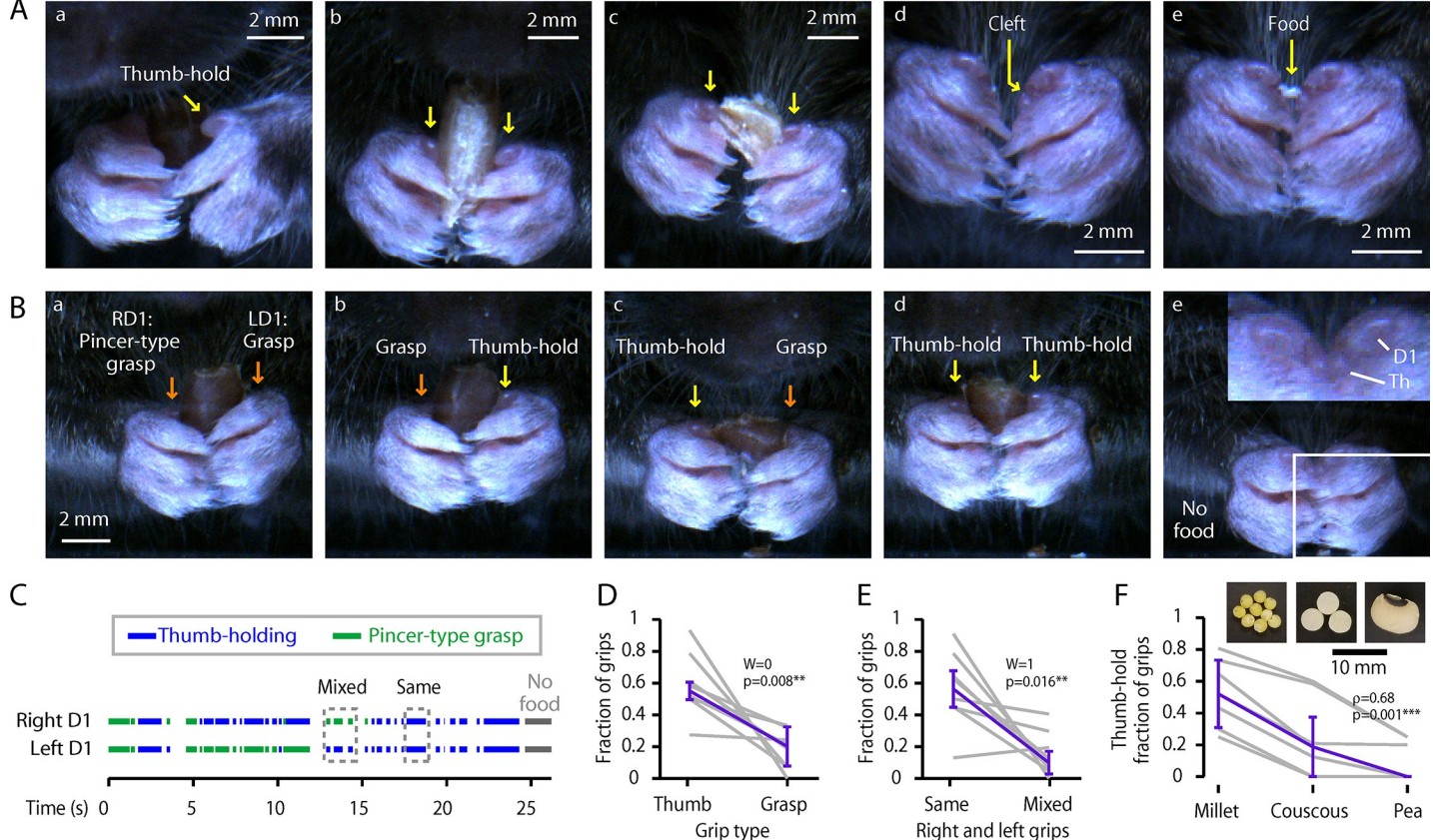

**Fig 2. Use of D1 for thumb-holding and pincer-type grips.** (A) Examples images showing use of D1 (arrows). (a) Thumb-holding, with left D1 apposed the side of a flaxseed (3/4 view). (b,c) Bilateral thumb-holding of a wheat berry. (d) Hands in food-holding posture, showing the thenar cleft. (e) A morsel of food held between the thumb clefts. (B) More examples of thumb-holding grips, and pincer-type grasping. Panel e shows a close-up of the empty cleft. (C) Example ethogram of grip types (green: pincer-type grasping; blue: thumb-holding). (D) Relative frequency of grip types. (Not shown: indeterminate grip types.) Gray lines: individual mice ($n$ = 8); purple: median ± m.a.d. across mice. (E) Relative frequency of bimanually symmetric (i.e., same on both hands) and asymmetric (mixed) grips. (Not shown: indeterminate grip types.) Gray lines: individual mice ($n$ = 8); purple: median ± m.a.d. across mice. (F) Fraction of grips that were thumb-holds for millet, couscous, and black-eyed peas, showing the dependence of grip type on seed size. Gray lines: medians for each mouse ($n$ = 7); purple: median ± m.a.d. across mice.

holding: 34 ± 12%; signed-rank versus flaxseed: $W$ = 1, $p$ = 0.016**; **S1A–S1C Fig, S3 Video**), suggesting that thumb-holding may be associated with smaller items. To test this possibility, we filmed and ethogrammed a second cohort of mice ($n$ = 7) eating millet (diameter ~2 mm), couscous (~4 mm), and black-eyed peas (~8 mm), which vary greatly in size but have similar shape. Thumb-holding proportion was significantly anti-correlated with seed size (Spearman's rank correlation coefficient: $n$ = 7 mice, 10 holding periods per mouse per seed type, $\rho$ = 0.68, $p$ = 0.001***; **Fig 2F**), confirming that mice preferentially use thumb-holding to manipulate smaller items–consistent with the observed use of only the two thumbs to hold the smallest items.

## Mice cycle rapidly between oromanual and holding phases of food-handling

We also noticed that feeding behavior appeared to consist of rapid alternation between two modes, or phases: an "oromanual" phase, bouts of active handling while gnawing on or bite-holding the morsel, and a "holding" phase, short pauses marked by relatively static holding posture (**Fig 3A; S4 Video**). Hence, we annotated the videos to label the oromanual and

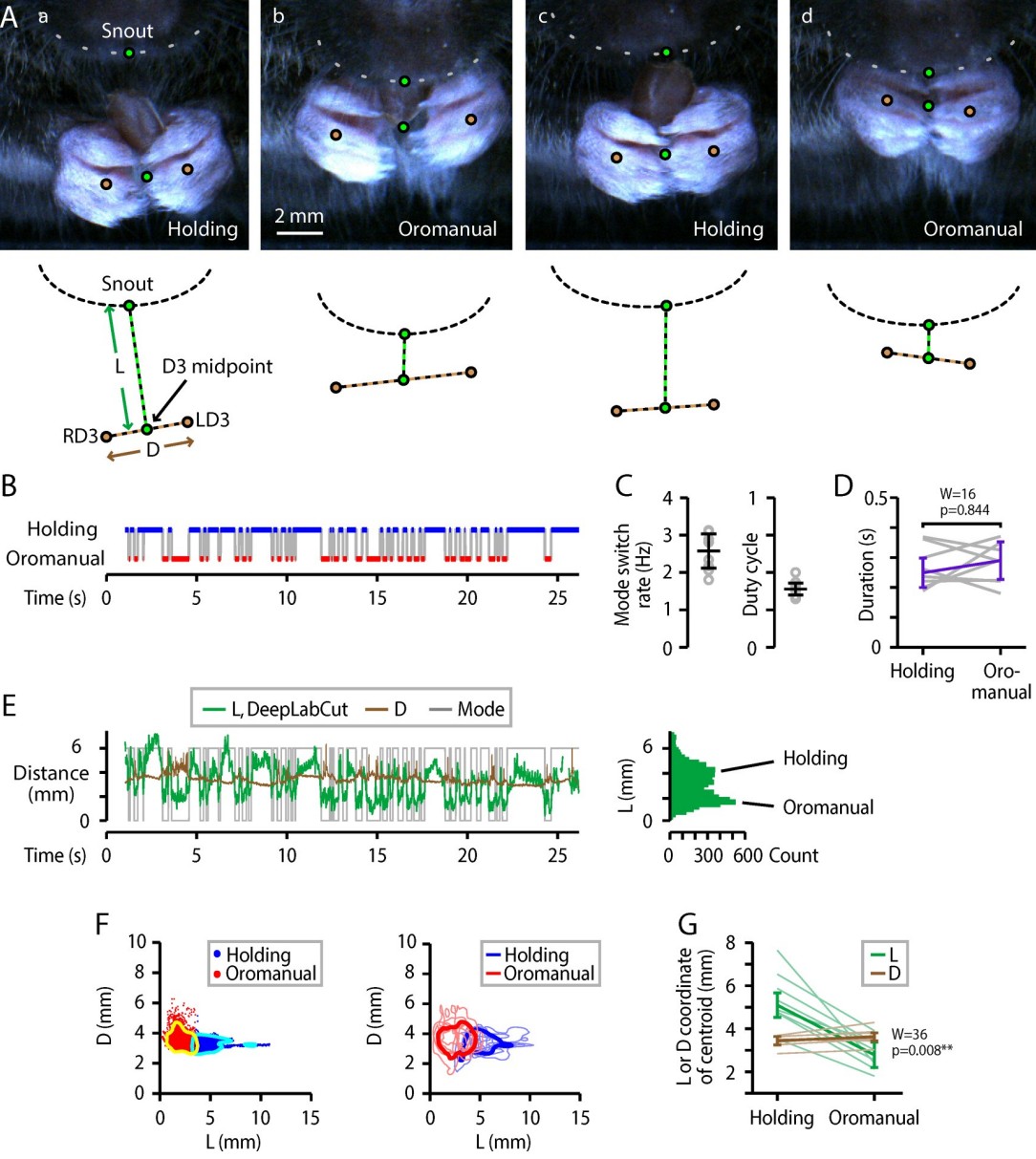

**Fig 3. Mice cycle rapidly between oromanual and holding phases of food-handling.** (A) Examples of holding (a,c) and oromanual (b,d) phases, for a mouse eating a flaxseed. Circles mark the snout-tip, right and left D3 digits (RD3, LD3), and midpoint between the D3s. Dashed lines represent the inter-D3 distance (*D*) and distance from snout-tip to D3 midpoint (*L*). Gray dashed lines: lower margin of the snout. (B) Ethogram of oromanual (red) and holding (blue) phases. (C) Left: Rate of switching between the two modes. Right: average oromanual duty cycle (total time spent in oromanual mode divided by total time in either mode). Circles: individual mice (*n* = 8); error bars: median ± m.a.d. (D) Average durations of holding and oromanual phases. Gray lines are individual mice, purple lines are median ± m.a.d. across animals. (E) Left: Plot of the D3-to-nose distance (*L*, green) measured by automated (DeepLabCut) tracking, along with the inter-D3 distance (*D*, brown) and ethogram (gray). Right: Histogram of *L*. (F) Left: *k*-means (*k* = 2) clustering of *L* and *D* for the single mouse example video shown in (E). Blue dots: holding; red dots: oromanual; cyan and yellow lines: 10% contour lines of 2D kernel smoothed density (see Methods) of the points in the respective clusters. Right: Average 2D kernel smoothed density of *L* and *D* for the holding (cyan) and oromanual (red) clusters. Light blue and red lines: 10% maximum density contour lines for the holding and oromanual clusters, respectively, for each mouse. Dark blue and red: 10% contour lines of the average kernel smoothed density across all mice. (G) *L*- and *D*-coordinates of the centroids of each cluster (i.e., the mean D3-to-nose distance or inter-D3 distance for points that fall into each cluster). Thin lines: individual mice; thick lines: median ± m.a.d. across mice.

holding phases. The resulting binary ethograms showed rapid cycling between modes (**Fig 3B**). The average frequency of mode-changes was 2.6 ± 0.5 Hz (median ± m.a.d., $n$ = 8 mice; **Fig 3C, left**). The duty cycle–the fraction of time spent in the oromanual phase–was 40 ± 3% (**Fig 3C, right**). Durations of both phases were brief, around 250–300 ms (**Fig 3D**). Results for wheat berries were similar, except for a slightly longer duty cycle of 52 ± 7% (signed-rank versus flaxseed: $n$ = 8, $W$ = 34, $p$ = 0.02*; **S1D Fig**).

For further quantitative characterization, we used DeepLabCut [16] to automatically track the positions of the digits and snout-tip. We considered that the distance between the digits and the mouth shrank during the oromanual phase, and that the hands often separated, with the food held stationary in the mouth. We therefore measured $D$, the inter-D3 distance, as a measure of hand separation, and $L$, the distance between the snout-tip and the midpoint between the D3s, as a measure of the hand-to-mouth distance (**Fig 3A**). Indeed, these distances varied systematically with the oromanual and holding phases (**Fig 3E**).

The oromanual and holding phases appeared to represent distinct states, and indeed $k$-means clustering on $L$ and $D$ revealed two clusters (**Fig 3F**). Clusters were mainly separated along the D3-to-nose dimension ($L$), with the cluster with smaller D3-to-nose and larger inter-D3 distance corresponding to the oromanual phase (**Fig 3F**). These results indicate that the oromanual and holding phases are distinct, with hands positioned ~2 mm lower during the holding phase (holding $L$ centroid: 5.1 ± 0.6 mm; oromanual $L$ centroid: 2.7 ± 0.6; signed rank: $n$ = 8, $W$ = 0, $p$ = 0.008**; **Fig 3G**). The $L$ and $D$ data clustered similarly for wheat berries (**S1E Fig**).

## Rapid "regrip" maneuver

Inspection of videos at extremely slow playback speeds revealed that, during oromanual phases, mice would often swiftly adjust their grip on the food item while holding it in the mouth. These bite-hold-assisted regrip maneuvers (**Fig 4A, S5 Video**) involved a transiently wider bimanual grip aperture (i.e., increased separation between the hands), and accordingly could be identified by sharp peaks in the $D$ trace occurring during the oromanual phase (**Fig 4B**). Peak-aligned regrips were strikingly similar (**Fig 4C and 4D**). Regrip maneuvers were extremely rapid, with average duration (full-width at half-maximum) of only 17 ± 0 ms (**Fig 4E**). Regripping occurred with an overall frequency of 1.5 ± 0.2 Hz ($n$ = 8 mice, **Fig 4E**). The average amplitude of peak-aligned regrips was 1.0 ± 0.1 mm (**Fig 4E**). Regrips tended to be repetitive, as reflected in an oscillatory component in the peak-aligned traces (**Fig 4C and 4D**) and supported by autocorrelogram analysis (**Fig 4D, inset**). The shoulders of the correlograms occurred at 65 ± 5 ms, corresponding to a periodicity of 15.5 ± 1.3 Hz. Regrips tended to involve both hands equally (i.e., bimanually symmetric) but were often biased to one or the other hand, and unimanual regrips were not uncommon (**Fig 4F**). Regrips were associated with dynamic changes in hand angle (**Fig 4G and 4H**). Regrips were similarly used for handling wheat berries (**S1F Fig, S3 Video**).

## Rapid "sniffing" maneuver

Inspection of videos with a ventral view of the hands and snout, captured by placing a mirror under the chamber ($n$ = 7 mice), revealed a sniffing-like maneuver: the mouse, while holding the food item between both hands, briefly brought it directly under the nose and nares (**Fig 5A, S6 Video**). We measured $L_{\text{ventral}}$, the distance from the snout-tip to the mid-point of the D3s, in these ventral views, which showed sharp dips coinciding in time with "sniffs" identified by visual inspection; like regrips, sniffs were readily identified by a simple peak detection algorithm (**Fig 5B**). The average amplitude of the sniffing movement was 2.1 ± 0.2 mm (**Fig 5E**)

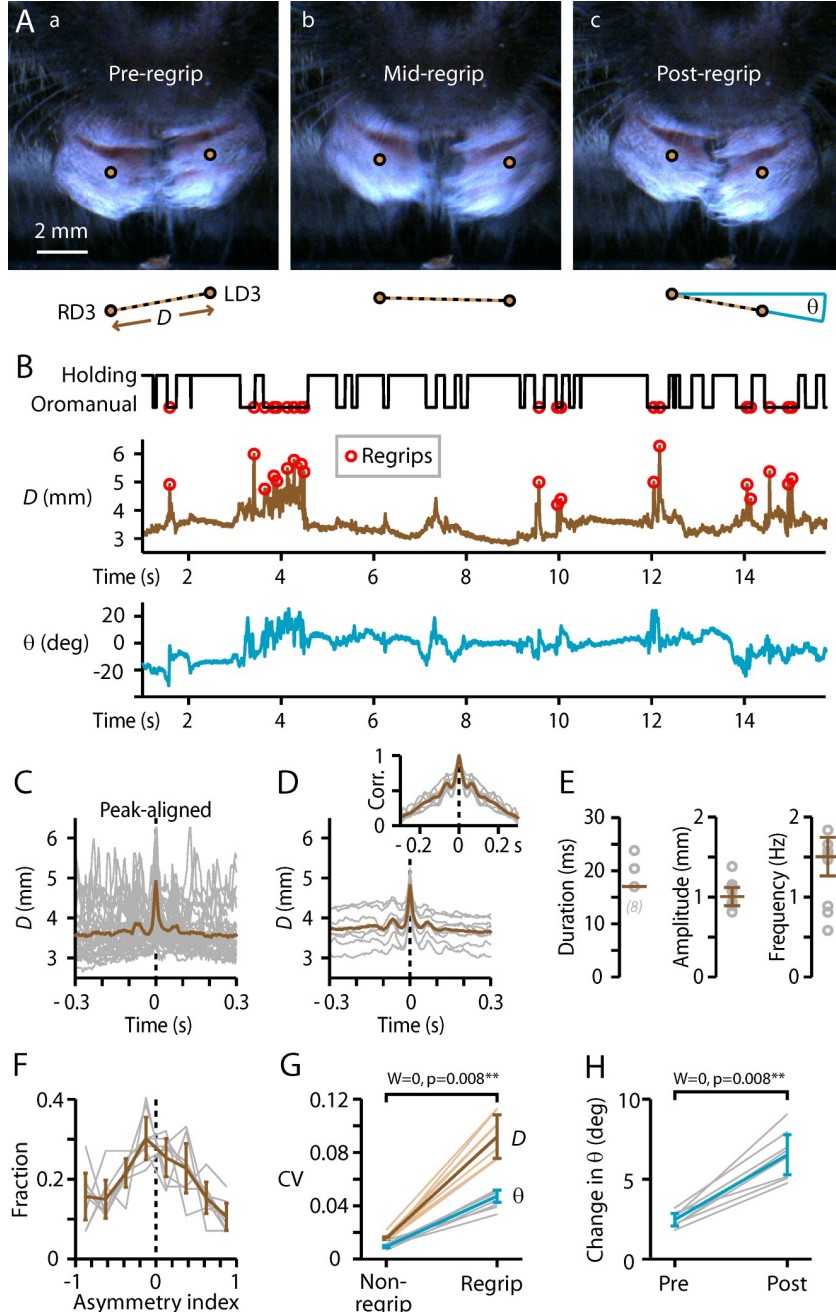

**Fig 4. Rapid "regrip" maneuver.** (A) Example frames from immediately before, during, and after a maneuver that involved bite-holding the item (flaxseed) while releasing its grip. (B) Top: ethogram, with regrips indicated by circles. Bottom: Plots of $D$ (brown), and $\theta$, the inter-D3 angle (blue). (C) Regripping events, peak-aligned (gray) and averaged (brown), for all videos from one mouse. (D) Same, but for the average traces from each mouse (gray, $n = 8$) and the overall average (brown). Inset: autocorrelograms (individual, gray; group average, brown). (E) Regrip parameters. Circles: individual mice; error bars: median ± m.a.d. (F) Histogram of regrip asymmetry index (see **Methods**). Gray: individual mice; brown: median ± m.a.d. (G) Rolling coefficient of variation of $D$ (brown) and $\theta$ (blue) during regrips versus non-regrip intervals (see Methods). Thin lines: individual mice; thick lines: median ± m.a.d. (H) Change in angle relative to baseline before and after regrips.

and the duration was 80 ± 16 ms (**Fig 5C–5E**). Sniffs occurred with an average frequency of 0.2 ± 0.2 Hz (**Fig 5E**). Sniffs tended to occur near the end of a holding phase, relatively soon

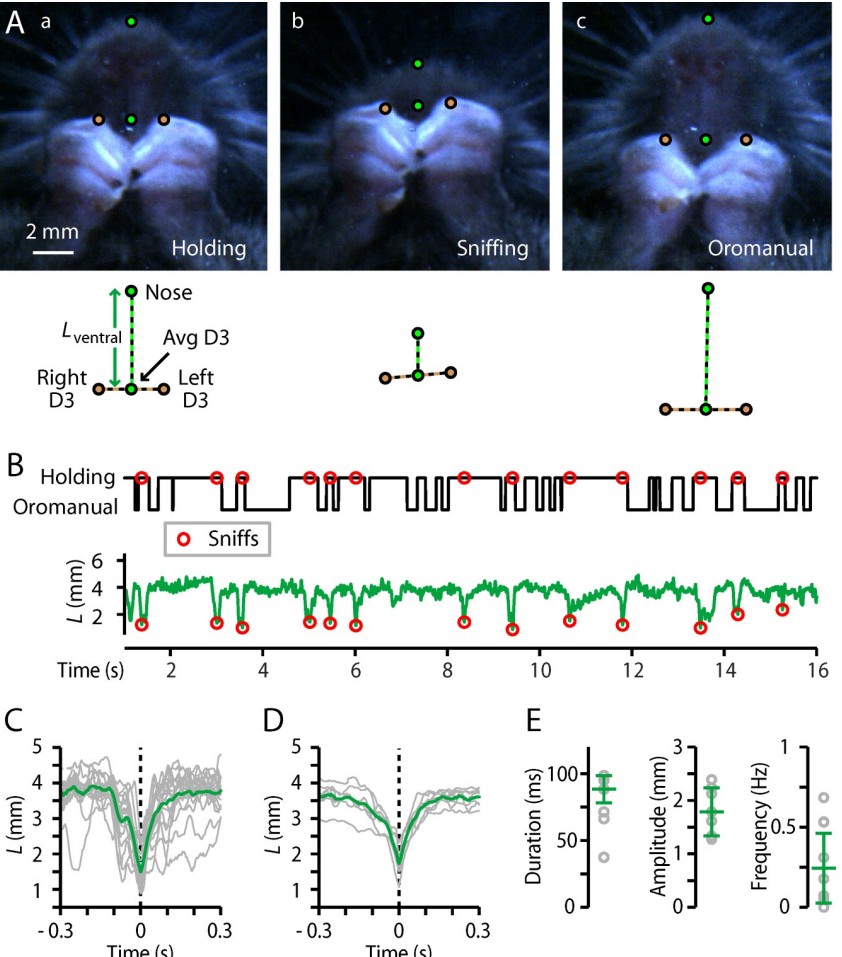

**Fig 5. Rapid "sniffing" maneuver.** (A) Example frames (ventral-view video) from immediately before (a), during (b), and after (c) a movement that put the food item under the nares. (B) Top: Ethogram with sniff events indicated by circles. Bottom: plot of $L_{ventral}$. (C) Sniff events, peak-aligned (gray) and averaged (green), for all videos from one mouse. (D) Same, but for the average traces from each mouse ($n = 7$ mice) and the overall average (green). (E) Sniff parameters. Gray circles: individual mice; error bars: median ± m.a.d.

($119 \pm 8$ ms) before switching to oromanual handling compared to the time since the last oromanual phase ($350 \pm 133$ ms; Wilcoxon's signed-rank: $W = 2$, $p = 0.047^*$; Fig 5B). Sniffs were also present for wheat berries (S1G Fig, S3 Video).

## Complex movements during oromanual handling

In addition to the set of common and stereotyped basic movements described above, we observed more complex manipulations during oromanual handling, also involving D1. Some of these movements generated rotations of the food item. For example, the D1s were used to roll the seed ~180˚, in coordination with other rapid movements of the hands and mouth (S7 Video). D1 and hand movements tended to be highly asymmetric during such rolling rotations.

We also observed double-gripping of two fragments, on rare occasions when seeds broke in two (S8 Video). To explore double-holding we gave mice couscous pellets, which have a greater tendency to break. We identified 20 instances of double-holding among $n = 13$ mice (6

from the main cohort and 7 from the seed size cohort). Two forms of double-holding were observed: in one, one fragment was held between the thumbs and the other pressed into the volar pads by D3-5; in another, one fragment was held in each hand with the two fragments pressed together. Regrips occurred during double-holds, without dropping of either fragment, and were usually unimanual, although small bimanual regrips were also seen. During unimanual regrips, the fragment that was not held in the mouth could be held by either the regripping or non-regripping hand (S8 Video, example 2). Diverse types of nimble digit movements and postures were observed during double-holding, such as regrips involving primarily D2, and use of D4-5 to bring the second fragment to the mouth after consuming the first fragment (S8 Video, example 3).

## Head-fixed mice display largely similar D1 and hand movements

As an initial step towards studying the neurobiology underlying dexterous food-handling, we trained mice to tolerate head-fixation while feeding on seeds (**Methods**), and analyzed videos (**Fig 6A and 6B**; **S2 Fig**; **S9 Video**). Head-fixed mice exhibited the same set of stereotyped movements, with a few relatively minor differences in certain parameters. Head-fixed mice used their D1s for both thumb-holding and pincer-type grasps (**Fig 6C, S9A Video**); rates of thumb-holding were similar (head-fixed thumb-holding: 54 ± 14%, median ± m.a.d., $n = 7$ mice; Mann-Whitney $U$ versus freely moving: $U = 76$, $p = 0.19$) but rates of pincer-type grasping were somewhat higher (43 ± 16%, $U = 44$, $p = 0.02^*$; **S2A and S2B Fig**). Cycling between distinct oromanual and holding phases was also preserved (**Fig 6B and 6D**; **S9B Video**), but slower (1.4 ± 0.3 Hz, $U = 92$, $p < 0.001^{***}$; **S2C Fig**). Regrip maneuvers resembled those of freely-moving mice (**Fig 6E, S9C Video**), with similar frequency (1.5 ± 0.4 Hz, $U = 51$, $p = 0.15$) and duration (20 ± 2 ms, $U = 57$, $p = 0.46$) but larger amplitude (1.5 ± 0.3 mm, $U = 43$, $p = 0.014^{**}$; **S2D–S2F Fig**). The previously identified sniffing maneuver was present (**Fig 6F, S9D Video**), with similar amplitude (2.3 ± 0.3 mm, $U = 35$, $p = 0.23$) and duration (60 ± 26 ms, $U = 46$, $p = 0.50$; **S2H and S2I Fig**), but significantly less frequent (0.04 ± 0.03 Hz, $U = 67$, $p = 0.008^{**}$; **S2G Fig**).

## D1 use by the squirrel during feeding

Squirrels are the only rodents previously noted to use bimanual thumb-holding to handle food [12]. We therefore assessed how D1 use by mice compared with squirrels. We recorded videos of wild gray squirrels (*Sciurus carolinensis*; $n = 2$) feeding on peanuts (unshelled), and analyzed these using a similar approach as for the mouse. As shown in the example (**Fig 7A**; **S10 Video**), the squirrel held the peanut between its D1s, using what appeared to be a bimanual thumb-holding grip (**Fig 7A**). Anatomical features of the squirrel's hand included small D1s bearing a flat (rather than claw-like) nail [12], larger thenar and hypothenar pads, and elongated D2-5 (**Fig 7B–7D**).

Similar to mice, squirrels alternated between holding and oromanual phases of food handling. Switching occurred at 1.1 + 0.1 Hz (median ± m.a.d., $n = 2$ squirrels), with an oromanual duty cycle of only 17 ± 3% (**Fig 7M**), much lower than that observed for mice eating any type of food, including peanuts (44 ± 3%, $n = 7$ mice). Oromanual phases were much briefer (0.25 ± 0.02 s) than holding phases (1.52 ± 0.02 s, **Fig 7N**). Phase transitions (holding to oromanual) were stereotypical and extremely fast (0.25 ± 0.02 s; **Fig 7O and 7P**).

Virtually all grips identified were thumb-holds (56/57), nearly all (52/56) of which involved no contact of D2-5 with the food (**Fig 7E–7G**). Rapid regripping maneuvers also occurred, in which the food item appeared to be gripped by the mouth while one or both hands released their grip, followed by a rapid readjustment of the grip (**Fig 7H–7K**; **S10 Video**, example 1).

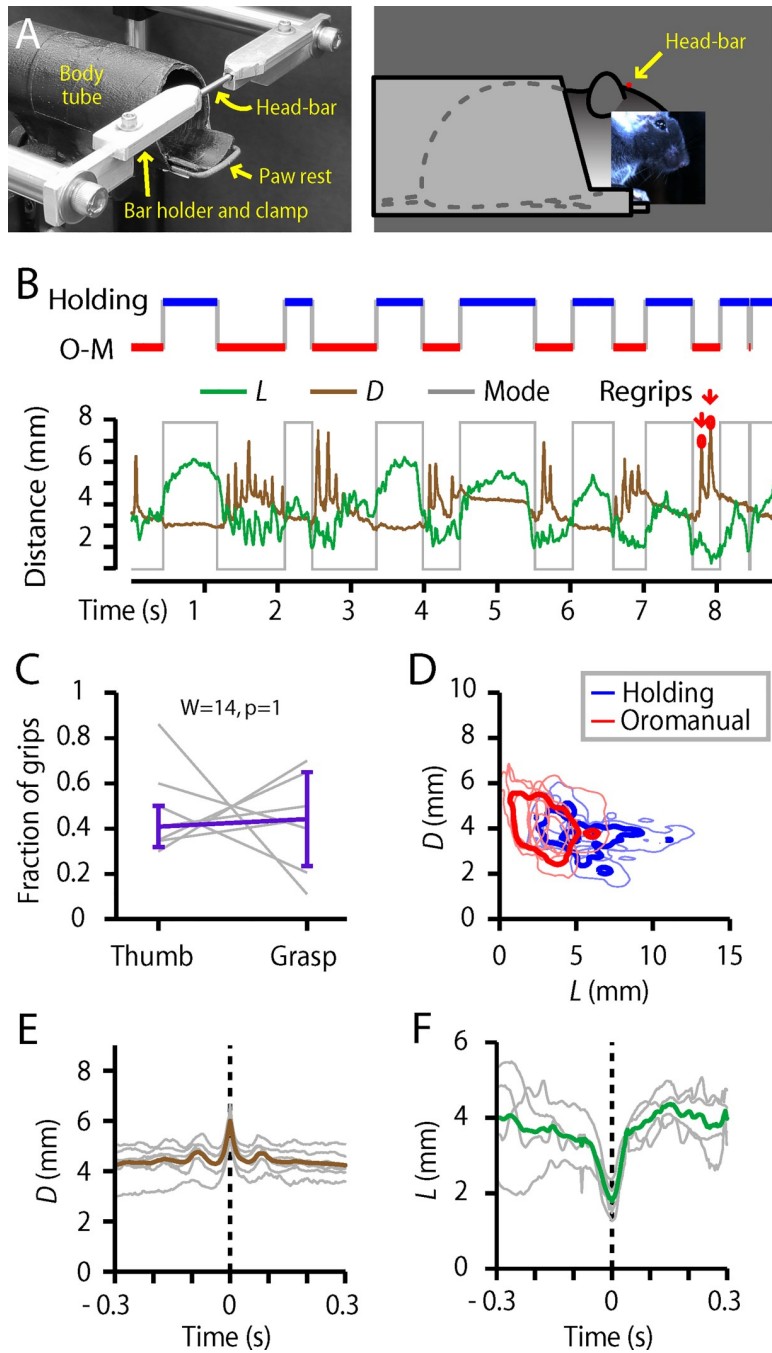

**Fig 6. Head-fixed mice display largely similar D1 and hand movements.** (A) Head-fixation apparatus. Left image shows a view with the head-bar (rod) in place. Right image depicts the head-fixed mouse in the body tube. (B) Top: Example ethogram. Bottom: Time series of *L* (green) and *D* (brown), along with the same ethogram (gray). (C) Relative frequency of grip types (*n* = 7). (Not shown: indeterminate grip types.) (D) Right: *L-D* clustering for all mice handling flaxseeds while head-fixed. Light blue and red lines show the 10% maximum density contours lines for the holding and oromanual clusters, respectively, for each of *n* = 7 mice. Dark blue and red show the 10% contour lines of the average kernel smoothed density across all mice. (E) Right: Average peak-aligned regrip traces from each mouse (gray, *n* = 7) and the overall average (brown). (F) Right: Average peak-aligned sniff traces from each mouse (gray, *n* = 4 mice) and the overall average (green).

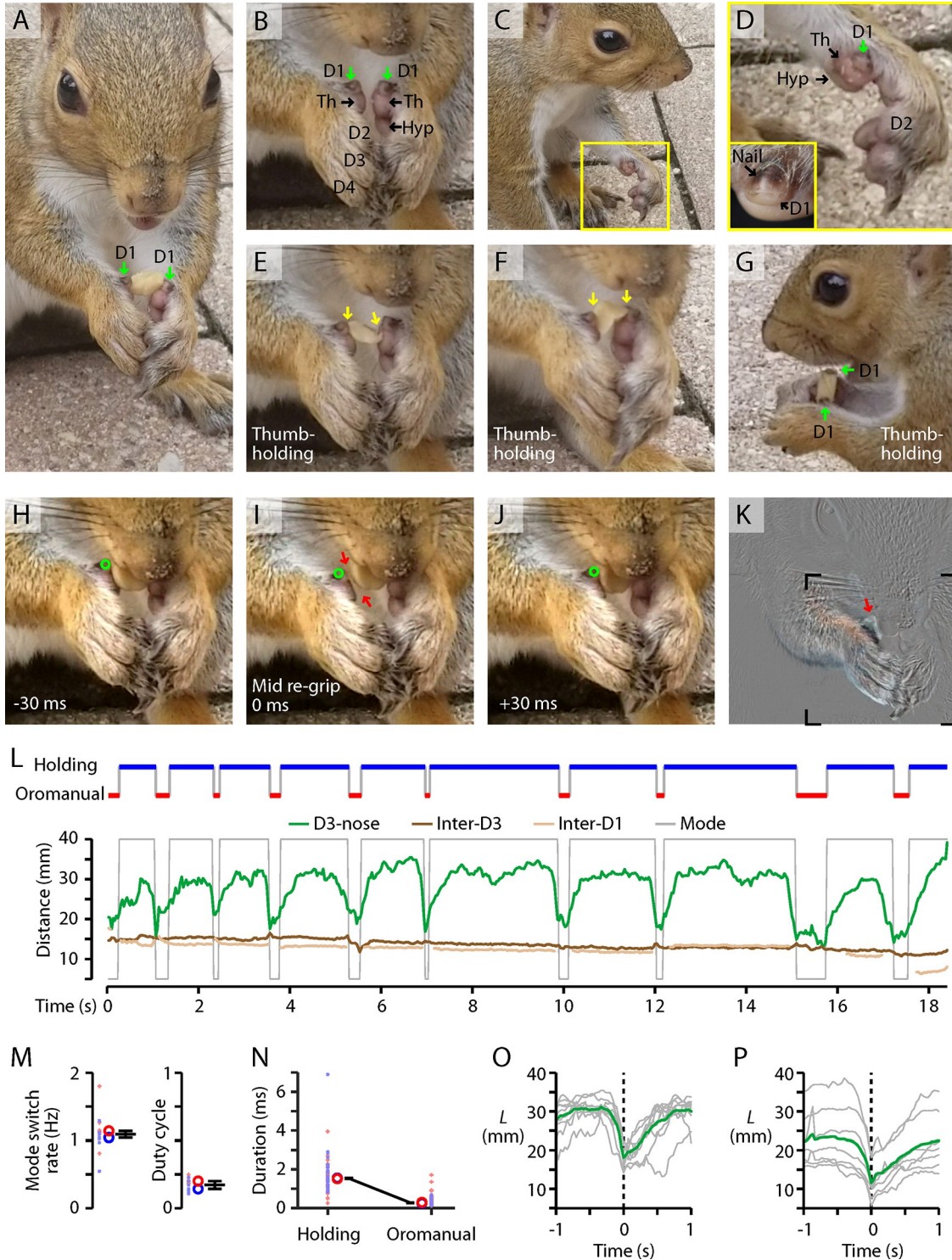

**Fig 7. D1 use by squirrels during feeding.** (A) Squirrel holding a peanut between its D1s, using a bimanual thumb-holding grip with no apparent contact by the other digits. (B) Closer view of the same squirrel's hands, after consuming the nut, showing the small D1s (green arrows), larger thenar (Th) pads, one hypothenar (Hyp) pad, and other digits (D2-4). (C, D) Side view, showing the elongation of D2 and other digits relative to D1 and the thenar and hypothenar pads. Close-up view (D, inset) shows the hard nail on D1 of a gray squirrel. (E, F) Example frames showing bimanual thumb-holding grip. Arrows mark where the morsel appears to contact the right hand at both the D1 and thenar pad, and the left hand at either the thenar pad (E) or D1 (F). (G) Side view, showing both D1s pressed against the side of the nut. (H-K) Regripping. Frames taken immediately before (H), during (I), and after (J) a rapid

regripping maneuver, during which the squirrel clamped the nut in its mouth and released its right thumb-holding grip. During the regrip (I), a small gap (arrows) opens between the right D1 (circle) and the nut. The difference image (K) of the mid-regrip versus preceding frame confirms that movement primarily involved the right hand and especially the D1 region. Boxed region in panel K indicates the region shown in the preceding panels. (L) Top: Ethogram of holding and oromanual handling phases, based on manual annotation of video frames. Bottom: The same ethogram (gray) is shown along with the automatically tracked D3-nose distance (green), and also the inter-D3 (brown) and inter-D1 (beige) distances. (M) Plots of the mode-switch rate (left) and oromanual duty cycle (right), for the $n = 2$ squirrels. Each squirrel's data is plotted with a different color and symbol. Small symbols: individual data for one video segment (one segment per peanut; $n = 13$ segments); large circles: median for each squirrel; error bars: median ± m.a.d. across squirrels. (N) Durations of the oromanual and holding phases, for the same animals. Small symbols: individual phases, large circles: median for each squirrel, error bars: median ± m.a.d. across squirrels. (O) $L$ traces aligned to the manually-identified switch from holding to the oromanual phase (gray) and average (green) for the single squirrel example segment shown in panel L. (P) Same, but showing the averages for each video segment from the same animal as in (O) (gray, $n = 7$ segments), along with the overall average for that animal (green).

Unlike mice, during regripping maneuvers the squirrel appeared to keep its distal digits (e.g. D3) clasped together, using them as a fulcrum (i.e., bellows-like hinge) while slightly separating the proximal D1/thenar pads to make the readjustment. One sniff-like maneuver was observed (**S10 Video**, example 2).

## Discussion

Dexterous handling of food items is an ethologically critical activity for mice and other tetrapods that use their forelimb digits for feeding [19, 22, 23]. Previous studies have focused mainly on the earlier stages of rodent feeding behavior, identifying distinct kinematic components associated with the reach, grasp, and withdraw phases of forelimb movements [17, 24, 25]. We focused on the later stage of food-handling, in an effort to elucidate the microstructural composition of manual dexterity as mice manipulate seeds, an ethologically relevant type of food. Using high-speed close-up video and automated tracking, we identified and quantitatively characterized multiple previously unknown kinematic elements of mouse food-handling.

First, we found that mice extensively use the D1 for food-handling. We identified two grip types involving D1: a thumb-holding grip, with D1 used to press the food item from the side; and, a pincer-type grasp, with D1 placed behind the object and D2 in front. Among rodents, pincer-type grasping may be common [12, 19] but bimanual thumb-holding was previously reported only for squirrels [12]. Interestingly, bimanual thumb-holding has also been described for aye-aye lemurs [26], which furthermore possess a 'pseudo-thumb' (an extension of a radial sesamoid bone replete with its own pad) thought to be involved in gripping [27]–an independent evolutionary adaptation that, we speculate, could represent shared functionality with the thenar pads of mice and squirrels. Thumb-holding was preferentially used to manipulate smaller food items, and for the tiniest morsels, mice used "pure" bilateral thumb-holds; i.e., the only digits holding the food were the thumbs (and/or thumb-clefts). This grip, with the two thumbs "bimanually opposed", thus appears to be employed particularly for high precision. Mice nearly always handled seeds bimanually, and the two hands usually, but not always, adopted the same grip type (thumb-holding or pincer-type grasp). Results from several studies of rats suggest that the symmetry/asymmetry of forelimb movements during food-handling reflects the particular geometry of the food (e.g. pasta) [10, 28–30].

Second, we found that mice cycled rapidly between an active oromanual phase and a more passive holding phase as they handled seeds. In the oromanual phase, mice gnawed at the morsel while actively manipulating it, using rapid movements of the hands and mouth. In the holding phase, mice continued chewing while holding the food item away from the mouth. Whereas "dexterous manipulation" for humans typically and indeed literally involves a dominant (usually right) hand, for seed-handling mice it relies on a coordinated three-way

interaction between the hands and orofacial apparatus. A future challenge is to understand how rapid, fine-scale movements of the tongue [31], teeth/jaws, lips, and other buccal musculature are orchestrated with those of the hands, and particularly the thumbs.

Third, we identified a "regripping" maneuver, in which mice rapidly readjusted their grip while stabilizing the food with the mouth. The speed of regrips is remarkable, only ~20 ms. Regrips were stereotypic, appearing as characteristic 'spikes' in the $D$ time series traces, and tended to occur in short bursts at a characteristic short interval of only ~65 ms. Thus, regrips appear to constitute a basis for more complex sequences of movements. The extreme brevity of regrips suggests that they are ballistic actions, not under neural feedback control. Nevertheless, although regrips were highly stereotyped, certain regrip parameters, such as the hand angle and asymmetry, displayed considerable variability, suggesting they are tuned to the task at hand–manipulating the seed according to its current shape, size, orientation, and so on.

Fourth, we identified a "sniffing" maneuver. In this characteristic movement, mice brought food to their noses briefly in a rapid sniffing maneuver. The sniff maneuver was fast and brief, often lasting only ~100 ms, suggesting it may correspond to one or two sniffs. This possibility is consistent with the idea that single sniffs contribute to olfactory perception [32], and with previously measured sniff frequencies for mice, which are generally <10 Hz [33]. Olfaction has been shown to be critical for guiding reaching movements of mice and rats [21, 34]. The occurrence of sniff maneuvers during the seed-handling stage implies an ongoing role of olfaction as mice manipulate and consume food.

Collectively, our quantitative analyses show that the complex feeding movements of mice can be decomposed into discrete kinematic elements, and that mice use their thumbs in a highly dexterous manner during food-handling. Our findings raise many questions to guide further research. How are the basic movement elements–grip types, postural modes, regrips, sniffs– chained together, or perhaps 'chunked', to generate sequences of purposeful action? Are the more complex types of food-handling, such as rotations and double-holding, also decomposable into simpler, quantifiable elements? What are the neural mechanisms in somatosensory and motor circuits that fine-tune and adaptively adjust the otherwise stereotyped movement elements? Are regrips and other ultra-fast features of movements essentially ballistic, or under sensory feedback control? More generally, to what extent do the kinematic elements represent low-level reflexes, learned habits, or high-level goal-oriented behaviors? How does handedness influence food-handling movements? Approaches that can be harnessed to address these questions include *in vivo* recordings of neuronal and muscular activity, as well as silencing/activation manipulations in sensorimotor circuits; computational modeling, similar to approaches developed to model rodent movement sequences at a more macroscopic level [18, 35]; and, perturbation-type paradigms, such as have been used for manipulandum-based studies of limb trajectories [15, 36].

Many neurological disorders–ranging widely from cerebrovascular and demyelinating disorders to spinal cord injury, peripheral neuropathies and more–impair dexterous movements, and rodent forelimb tasks have become widely used as translational models to investigate neural mechanisms and therapeutic targets (e.g. [8, 37, 38]). High-speed close-up video of mice handling food has potential for further development as a quantitative assay to investigate how specific aspects of D1-related manual dexterity relate to neural circuit dysfunction in a wide range of neurological disorders.

## Materials and methods

### Animals

This study used experimentally naïve wild-type C57BL/6 mice (The Jackson Laboratory, Bar Harbor, Maine). At the time of video recordings, they were 58–187 days postnatal in age, their

starting body weights were 17.3–32.2 g (median ± m.a.d.: 25.3 ± 4.7 g), and their post food-restriction body weights were 14.3–28.1 g (median ± m.a.d.: 20.8 ± 3.8 g). Mice were monitored throughout the food restriction period for signs of ill-health [39], and body condition scores [40] were taken each day. No signs of ill-health were observed and no mouse fell below a score of 3 throughout the study. Mice of both sexes were used, consistent with NIH policy on sex as a biological variable in basic research. Sex-dependent differences were not anticipated, and for the major food-handling parameters relating to thumb-holding, mode-switching, regrips, and sniffs we did not find any obvious trends to suggest sex-dependent differences, i.e. data distributions for the two sexes were completely overlapping. We also did not observe any correlations with age (e.g. across 14 parameters for the main freely-moving cohort, all $R^2$ values were less than 0.40 and $p$-values were greater than 0.13) or weight (all $R^2 < 0.42$, $p > 0.08$).

Mice were bred in-house, housed in groups with a 12 hr light/dark cycle and free access to food and water until the time of food restriction (see below). Six mice (2 female, 4 male) were used for the main freely moving seed-handling study, seven (2 female, 5 male) for the analysis of seed size, and seven (5 female, 2 male) for the head-fixed seed-handling study. Two further mice (2 male) were mounted with head-fixation rods but failed to acclimatize to eating while head-fixed and so were used to augment the dataset for the freely moving study. Mice were used as they became available, and not randomized to each cohort or balanced by sex. All studies of mice were approved by the Northwestern University IACUC and fully complied with the animal welfare guidelines of the National Institutes of Health and Society for Neuroscience.

Video recordings of wild squirrels were made with the approval of and in full compliance with the regulations of the Division of Wildlife Resources of the State of Illinois Department of Natural Resources; a permit was not required because the study was purely observational and did not involve animal handling, protected species, or conservation land. The squirrels studied appeared to be adults based on their body size, but their age and sex were unknown. A roadkill squirrel's forelimb was photographed for close-up views of the D1.

### Feeding procedure

Mice were food restricted prior to the start of experiments to motivate feeding behavior. Mice were fed a measured amount of standard rodent diet each day to maintain their weight between 85 and 90% of pre-restriction body weight. On experimental days, food items were presented to mice, including flaxseeds (*Linum usitatissimum*; also called linseed), wheat berries (*Triticum aestivum*), hulled millet (*Panicum miliaceum*), couscous, dried black-eyed peas (*Vigna unguiculata*), and peanuts (*Arachis hypogaea*).

### Freely moving experiments

For experiments with freely moving mice, the food restriction period was generally 1–3 days before filming. During this time, mice were acclimatized to handling by the experimenter and the filming chamber. The filming chamber for freely moving mice was constructed from a large vertically oriented tube (3" inner diameter clear acrylic cylinder). On one side of the tube, to view the mouse clearly we placed a flat clear window (a 2¼" wide by 2" high acrylic panel). The chamber was mounted above a flat mirror angled at 45º to image the frontal and ventral aspects of the mouse simultaneously. Each mouse was imaged in one or two behavioral sessions. At the beginning of each session, mice were allowed to explore the chamber for ~30 min before feeding. Then, food items were placed one-by-one (or 2–4 at a time, for millet) in front of the filming window and videos taken as they consumed the seed. Mice were fed seeds until a sufficient quantity of high-quality (i.e., the hands, snout, and seed were in-focus and in-frame for the majority of the video, with lighting sufficient to distinguish the digits, and the rostro-

caudal axis of the mouse not more than 45º from the camera's line of sight) video was captured. This took 23–62 flaxseeds (30.5 ± 7 seeds, median ± m.a.d., *n* = 8 mice) and 1 or 2 wheat berries per mouse.

## Head-fixation experiments

Mice were food-restricted and familiarized with the experimenter and head-fixation apparatus following standard procedures [39], starting three days after surgery to implant a head-fixation rod (see **Head-fixation surgery** below). Briefly, mice were first acclimatized to handling by the experimenter, then introduced to the experimental chamber (a 3" long, 1¼" inner-diameter plastic tube with a ½" shelf protruding from one end; **Fig 6A**) and head-fixation apparatus, then to gentle head-fixation by hand-holding the head-fixation rod, then finally to full head-fixation. Mice were provided treats (Fruit and Veggie Medley; Bio-Serv, Flemington, NJ) as reward and encouragement during this process. Mice generally took ~3 weeks to become comfortable eating while head-fixed (22 ± 2.7 days, median ± m.a.d, *n* = 7 mice, range = 17–23). The holder for the head-fixation rod was designed so that the rod could be clamped either fully, preventing all head movement, or partially, allowing rotation of the head in the sagittal plane. In the main text and figures only the results for fully head-fixed mice are presented; results with partial head-fixation are discussed below. To film side views, either one mirror was placed at 45˚ to the side of the mice, or two mirrors and a prism (30 mm equilateral prism, #47–278, Edmund Optics, Barrington, NJ) were placed in front of the mouse to image front and side views with identical optical path lengths. To film ventral views, a single mirror was placed at 45˚ below the front of the mouse tube. On filming days, mice were head-fixed, placed in front of the camera, and presented with seeds while filming until at least 30 s of seed-handling behavior for each seed type (flaxseed, wheat berry), viewing angle (side, ventral), and head-fixation condition (fully-fixed, freely-rotating) had been captured. This took 12–26 flaxseeds (median ± m.a.d.: 17 ± 3.7) and 4–8 wheat berries per mouse (median ± m.a.d.: 5 ± 1.1). One mouse did not eat seeds with the ventral mirror in place, but consumed 4 flaxseeds and 3 wheat berries while head-fixed during filming from the front and side.

In mouse head-fixation experiments the head is typically horizontal, but mice are more hunched when feeding. We used partial head-fixation to assess how head-angle may affect food-handling movements. We calculated the head angle, Θ, by tracking the eye from the side (**S3A and S3B Fig**). Partially head-fixed mice adopted a nearly vertical head angle of -76 ± 2˚ (*n* = 7 mice, **S3C Fig**). In our experiments with fully head-fixed mice, the head was fixed at -67 ± 12˚. There were no major differences in seed-handling behavior between full and partial head-fixation.

## Head-fixation surgery

Under deep isoflurane anesthesia, mice were placed in a stereotaxic frame and a ~1 sq. cm. circular incision was made to expose the cranium. The periosteum was removed and the skull gently cleaned with hydrogen peroxide (Fisher Scientific, Rochester, NY). A stainless steel head-fixation rod (dowel pin, 316 stainless steel, 3/32" diameter, 7/8" long; McMaster-Carr, Elmhurst, IL) was placed on top of lambda, perpendicular to the central suture, and affixed using cyanoacrylate glue and dental cement (Ortho-Jet Liquid & Jet Denture Repair Powder; Lang Dental Manufacturing, Wheeling, IL). Mice were given 0.3 mg/kg buprenorphine preoperatively and 1.5 mg/kg meloxicam postoperatively as analgesia, followed by a second dose of meloxicam 24 hours after surgery.

## Videography

Videos were obtained with a CMOS-based color video camera (FLIR CM3-U3-13Y3C-CS; FLIR Integrated Imaging Solutions, Richmond, BC, Canada). Videos were acquired at 294.1 frames per second (fps), 3.3 ms exposure time, 18.1 dB gain, and 1280 by 512 pixel field of view. Direct and mirror-reflected images of the mouse were acquired simultaneously with a single camera (rotated 90˚ in the case of simultaneous front and ventral views), except for the seed size dataset where only the front view was captured. A zoom lens (A4Z2812CS-MPIR, Computar, Cary, NC) was mounted on the camera body, and a ring of white LEDs (LED-64S, AmScope, Irvine, CA) was mounted around the lens. For some experiments with freely moving mice, to magnify the image of the mouse's hands, a lens (planoconvex; focal distance, 130 mm; diameter, 92 mm) was mounted immediately in front of the LED ring. Video was captured using SpinView camera recording software (FLIR) as uncompressed AVI video directly to hard disk. Video segments and individual frames were adjusted with ffmpeg (ffmpeg.org) to improve brightness and contrast for display purposes, but raw videos were used for all analyses.

## Micro-CT

A mouse underwent transcardial perfusion-fixation with 4% paraformaldehyde following standard methods. The right upper extremity was amputated at the shoulder. The hand and distal forearm were imaged with at an isotropic voxel size of 0.063 mm (Mediso nanoScan). Image data were stored in DICOM format and converted to STL format for rendering.

## Ethograms

Segments of high-quality video (according to the criteria described above) were identified, extracted, cropped to 512 by 512 pixels, and rotated for upright viewing of the mouse using ffmpeg. These videos were imported into FIJI [41, 42] and reviewed frame-by-frame. Videos were divided into alternating oromanual and holding phases, where the oromanual phase was defined by any manipulation of the seed with the digits or mouth (including biting) and the holding phase was defined by periods of negligible digit movement while the mouse engages in other behaviors such as chewing or sniffing the environment. For each segment, the start frame and the type of phase was noted in a separate text file for each video. The start of the oromanual phase was defined as the frame in which the mouse appeared to bite down on the seed or began digit movement. The start of the holding phase was defined as the frame in which the mouse released the seed into the hands and began to lower the seed away from the mouth with the hands, and/or digit movement came to an end. For each holding phase, the grip type used by each hand was notated as thumb-holding if the thumb appeared to be apposed to the side of the seed (regardless of position of or contact by other digits), pincer-type grasping if the thumb was behind it, or ambiguous if the position of the thumb relative to the seed could not be discerned. We did not attempt to distinguish sub-types of pincer-type grasp. Phases (holding or oromanual) at the beginning and end of video segments were excluded from analyses of phase duration or switching frequency, as their true durations were unknown. Rather than ethogram the entirety of every video, for freely moving mice we ethogrammed until at least 10 holding phases with unambiguous grip types were identified for each mouse and food type, and from both the beginning and end of seed consumption. No systematic differences were found between seed-handling behavior at the beginning versus end of consumption, hence for the seed size and head-fixed cohorts we only ethogrammed until we identified 10 holding phases with identified grip types for each mouse, food type, and head-fixation condition. In initial studies, two observers were involved in developing the approach for

generating ethograms. This approach, once standardized (as described above), yielded essentially identical ethograms generated independently by two different observers. Thereafter, one observer (J.M.B.) generated all the ethograms used for analysis.

## Tracking

DeepLabCut [16] was used for markerless tracking. In front views, we tracked the tip of the snout, both D1s, and the knuckles (proximal inter-phalangeal joints) of D2-4 on both hands. In side and ventral views, only the nose and D3s were tracked. DeepLabCut was trained separately for each view with data from all 15 mice in the freely-moving and head-fixed cohorts for the front and ventral views, and from all 7 head-fixed mice for the side view. DeepLabCut was also trained separately for each individual squirrel. We used 150–400 frames for training each time. Videos were cropped to exclude irrelevant portions (usually to 512 by 512 pixels, except for side views, which used 400 by 512 pixels, and front views of four mice from the head-fixed cohort, which were cropped to 256 by 256 pixels), then anatomical features of interest were marked using the Multi-Point Tool in FIJI and exported to a comma-separated value (CSV) file. This CSV file was converted to the HDF5 format expected by DeepLabCut via a custom routine written in Python. DeepLabCut using the ResNet-50 neural network was trained on the annotated images for 1,030,000 iterations, then used to track the locations of the nose and digits in the full set of video segments. Frames with poor tracking were visually identified by manual inspection of the videos or $L$ and $D$ traces (see below), corrected using DeepLabCut's refinement GUI [43], and the model retrained until satisfactory tracking results were obtained. Sections of tracking that remained poor after refinement (i.e., exhibited large single-frame jumps or jitter) were excluded from analysis. To convert distances from pixel to millimeter units we length-calibrated the videos based on images of a ruler obtained during each recording session. All DeepLabCut-tracked body part trajectories were then imported into Matlab (The MathWorks, Natick, Massachusetts) for further analysis. For one video we compared manual and automated tracking, finding essentially identical results. For the video shown in Fig 1, the 7,453 X and Y co-ordinates measured by automated (DeepLabCut) versus manual tracking were highly correlated (X: $R^2 = 0.94$, $p < 0.0001^{***}$, Y: $R^2 = 0.96$, $p < 0.0001^{***}$).

## Trajectory analysis

To recenter and reduce the dimensionality of our dataset, we focused on two main features extracted from the DeepLabCut-extracted front-view body part trajectories: $D$, defined as the Euclidean distance between the right and left D3 locations, and $L$, defined as the Euclidean distance between the snout location and $D_{avg}$, the midpoint of the two D3 locations. This choice was guided by pragmatism rather than standard dimensionality reduction techniques: the nose provides an easily trackable locus to which to anchor the digit locations with respect to the mouse, and due to occlusions and pro/supination of the hands other digits were frequently untrackable. However, when we performed principal components analysis of the pairwise distances between every tracked body part, the two most explanatory dimensions for each video were usually driven primarily by the distances between each digit and the nose, and the distances between contralateral digits, thus corresponding roughly to $L$ and $D$. We also calculated $L$ in the side and ventral views: these are referred to as $L_{side}$ and $L_{ventral}$, respectively.

## Cluster analysis

Each video frame defines a point in $L$-$D$ space. The data from all video segments for each mouse eating each seed type were clustered using $k$-means clustering, varying $k$ from 1 to 10 with 10 replicates. The optimal number of clusters was found using the "L" method to find the

value of $k$ where adding additional clusters causes minimal improvement in the sum of squared distances between each point and the centroid of its cluster [44]. To visualize the density of points in each cluster, smoothed 2D density histograms of each cluster for each mouse were computed using the ksdensity function in the Matlab Statistics and Machine Learning Toolbox with a support of [0,20] mm in each dimension, covering the entire range of observed $L$ and $D$ values. Average kernel smoothed densities were calculated by interpolating individual mice's histograms over identical grids, normalizing to a maximum density of 1, and taking the mean over mice of the result. For individual mouse kernel smoothed densities and the average kernel smoothed density, the 10% contour line was taken as the contour line at 10% of the maximum density.

## Identification of sniffs and regrips

Sniffs and regrips could be identified by downgoing spikes in $L_{\text{ventral}}$ or upgoing spikes in $D$, respectively. Accordingly, regrips were detected using the findpeaks function in the Matlab Signal Processing Toolbox as peaks in $D$ with minimum prominence of 0.75 mm and a slope that exceeded 88 mm·s⁻¹ in either direction. Sniffs were detected in the additive inverse of $L_{\text{ventral}}$ with a minimum peak height of -2.5 mm, a minimum peak prominence of 1 mm, and a minimum peak separation of 250 ms. The 200 ms of video surrounding each detected sniff were extracted and manually inspected; instances where the mouse did not bring the seed under the nares were excluded. The peak detection parameters used for detecting freely moving sniffs produced false positives in the head-fixed dataset, so for those videos the minimum peak prominence was increased to 2.5 mm and the maximum peak width was set to 600 ms. For either maneuver, the peak height was defined as the absolute difference between the value of $L$ or $D$ at the peak and the mean value during the pre- and post-maneuver baselines, which were taken from 300 ms to 100 ms before and 100 ms to 300 ms after the maneuver. The full-width at half maximum (FWHM) was defined as the width of the peak at halfway between baseline and the peak value. Peak parameters were extracted for individual peaks, before averaging first within and then across mice.

## Analysis of regrips

To assess the periodicity of the regrip maneuver, groups of regrips with an inter-regrip interval of ≤ 300 ms were identified and the autocorrelogram of the corresponding $D$ trace computed from 300 ms before the first regrip of the group to 300 ms after the last regrip of the group, to a maximum correlogram lag of 300 ms. The mean of all such autocorrelograms, normalized to have a minimum of 0 and a maximum of 1, was taken for each mouse, and the regrip periodicity calculated as the reciprocal of the location of the largest peak of the mean autocorrelogram to the right of the central (zero-lag) peak.

To measure regrip asymmetry, for each regrip we calculated the two-dimensional path integral of each hand's trajectory during the 40 ms period centered on the peak of the regrip (40 ms was chosen as roughly twice the average regrip FWHM). We then calculated an asymmetry index for each regrip as the difference between the right and left hands' path integrals divided by their sum; the index ranges from 0 when both hands travel the same distance to 1 when one hand is stationary. The sign indicates the direction of the bias: negative to the left hand, and positive the right.

The variability of $D$ and the inter-D3 angle $\theta$ was measured using the rolling coefficient of variation, calculated as the ratio of the standard deviation of $D$ or $\theta$ during a 40 ms sliding window to the mean during the same window. Finally, to analyze changes in $\theta$ as a result of regrips, we took the median value of $\theta$ from 60 ms to 20 ms before the peak each regrip as the

baseline, and calculated the median absolute difference in angle from this baseline during the baseline epoch and the post-regrip epoch of 20 ms to 60 ms after the peak.

### Head-angle estimation

For the head-fixed seed-eating, mice were filmed from the side as well as the front, allowing us to measure the angle of the head. To do so, the eye on the imaged side was tracked using DeepLabCut. During full head-fixation the eye remains mostly stationary, but when the head is allowed to rotate freely the eye sweeps out an arc as the mouse rotates its head. The center and radius of the corresponding circle were calculated using Pratt's method [45] and the head angle Θ estimated as the four-quadrant inverse tangent of the displacement of the eye coordinate from the center of this circle. Depending on which head-fixed filming setup was used, videos were flipped to ensure the eye appeared on the left side of the image relative to the center of the circle.

### Phase transition rise time analysis

To quantify duration of phase transitions from holding to oromanual in the squirrel, for each phase transition a baseline was calculated as the mean value of $L$ from 750 to 250 ms before the phase transition. A line was then fit passing through 10% and 90% of the difference between this baseline and the value of $L$ at the time of the phase transition. The duration of the phase transition was then taken as the time before the phase transition at which this line crossed the baseline.

### Statistical analysis

In order to minimize assumptions about the underlying distributions of the data, non-parametric statistical tests were used throughout. Values of $n$ for each test represent the number of mice unless otherwise specified and are given in the Results section, figures, and figure legends. Sample sizes were not calculated *a priori*, but at least 6 mice were used for each cohort as the non-parametric tests used are underpowered to detect differences of any size for samples smaller than this. As the majority of analyses in this paper were automated, blinding was not necessary. Mice of both sexes were used (13 male, 9 female); the small sample size within each cohort precludes between-sex comparisons. All mice from the relevant cohorts were included in all tests, except for analyses of sniffing frequency, where mice without ventral view data were excluded, and analyses of sniff parameters, where mice with no detected sniffs were excluded.

In the main text, group data are presented as median ± median absolute deviation (m.a.d.) unless indicated otherwise. In the figures, data are plotted as the mean along with individual traces for time series data, and as median ± median absolute deviation (m.a.d.) for all other data, unless otherwise specified. Where appropriate, averages (medians or means, as indicated) are taken first within and then between animals. Medians from independent samples were compared using the Mann-Whitney $U$ test, medians from paired samples were compared using Wilcoxon's signed-rank test. Spearman's rank-correlation coefficient was used for all correlations except between manual and DeepLabCut-based data, which were compared with Pearson's product-moment correlation coefficient. All $p$-values less than 0.05 were considered significant and are highlighted in the text and figures with *. The $p$-values that remained significant after false discovery rate correction for multiple comparisons [46] are highlighted with ** and after Bonferroni correction with ***.

## Data and code availability

The tracking data and manually generated ethograms are available on SourceForge at https://sourceforge.net/p/seed-handling-paper/code, along with the Python and Matlab code used to analyze the data presented in this paper. The videos and trained DeepLabCut models used to track digit locations are available on Zenodo at https://zenodo.org/record/3531964.

## Supporting information

**S1 Fig. Characteristic D1 and hand movements generalize to another seed type (wheat berry).** (A) Top: Example oromanual/holding ethogram for a mouse feeding on a wheat berry. Bottom: Time series of *L,* the distance between the nose and the average D3 position (green), and *D*, the distance between the D3 digits (brown). The same ethogram is also shown (gray). (B) Left: Relative frequencies of grip types; indeterminate hold types are not included. Gray lines: individual mice ($n = 8$), error bars: median ± m.a.d. across mice. The lower rate of thumb-holds was the main difference compared to flaxseed (see text). Right: Relative frequency of symmetric (i.e., same on both hands) and asymmetric grip types as a fraction of all grip types. (Not shown: indeterminate grip types.)
(C) Fraction of thumb-holds (left) and symmetric grips (right) for flaxseed versus wheat berries.
(D) Mode switch rate (left) and duty cycle (right) for flaxseed versus wheat berries.
(E) Left: Example of *k*-means clustering ($k = 2$) of *L* and *D*, for the video shown in (A). Blue dots correspond to the holding phase and red the oromanual phase. Cyan and yellow lines show the 10% contour line of 2D kernel smoothed density of the points in the respective clusters. Right: *L-D* clustering for all mice. Light blue and red lines show the 10% maximum density contours lines for the holding and oromanual clusters, respectively, for each of $n = 8$ mice. Dark blue and red show the 10% contour lines of the average kernel smoothed density across all mice.
(F) Left: Peak-aligned individual (gray) and average (brown) regripping events for all wheat-handling videos from one mouse. Right: Average regrip traces from each mouse (gray, $n = 8$) and the overall average (brown).
(G) Left: Peak-aligned individual (gray) and average (green) sniffing events for all wheat-handling videos from one mouse. Right: Average sniff traces from each mouse (gray, $n = 6$ mice) and the overall average (green).
(PDF)

**S2 Fig. Comparison of freely moving and head-fixed behavior.** Gray circles: individual mice; error bars: median ± m.a.d. (A) Fraction of thumb-hold grips.
(B) Fraction of pincer-type grasps.
(C) Mode switch rate.
(D) Regrip frequency.
(E) Regrip amplitude.
(F) Regrip duration.
(G) Sniff frequency. Mice with insufficient ventral view data excluded.
(H) Sniff amplitude. Mice with no detected sniffs excluded.
(I) Sniff duration. Mice with no detected sniffs excluded.
(PDF)

**S3 Fig. Analysis of head angle with partially head-fixed mice.** (A) Left: Side-view images of a head-fixed mouse at rest (left) and during the holding (middle) and oromanual (right) phases. Purple dashed line shows the arc swept by the eye as the head rotates about the head-fixation

rod. Right: Measurement of head angle (Θ). An example is shown of the individual eye locations measured in the video (purple dots), with the center and circumference (gray) of the circle that best fits these points according to Pratt's method (see Methods).

(B) Top: Example ethogram of handling behavior for a partially head-fixed mouse feeding on a flaxseed, based on manual annotation of holding (blue) versus oromanual (red) phases. Bottom: Time series of the tracked $L_{3D}$, the three-dimensional distance between the nose and the average D3 position (green trace); $D$, the distance between the D3 digits (brown trace); and Θ the angle of the eye about the center of rotation (purple trace).

(C) Head angle Θ during holding versus oromanual phases for partial head-fixation. Thin lines are individual mice, error bars are median ± m.a.d over all mice ($n = 7$).
(PDF)

**S1 Video. Example of a mouse feeding on seed.** An example of a mouse feeding on a flaxseed, at actual speed (1x speed), then repeated at 0.1x speed.
(MP4)

**S2 Video. Use of D1 for thumb-holding and pincer-type grips, and views of the thumb cleft.** (A) An example of a mouse switching between thumb-hold, mixed, and pincer-type grasps shown at actual speed, then repeated at 0.1x speed with the different grip types annotated.
(B) Three brief video segments in which the thumb and thenar pad are visible, with pauses to highlight the thumb cleft.
(MP4)

**S3 Video. Characteristic D1 and hand movements generalize to another seed type (wheat berry).** (A) An example of a mouse holding a wheat berry in various grips shown at actual speed, then repeated at 0.1x with the different grip types annotated.
(B) An example of a mouse feeding on a wheat berry at actual speed, then repeated at 0.1x speed the oromanual and holding phases annotated.
(C) An example of a brief period of oromanual handling of a wheat berry including several regrip maneuvers shown at actual speed, then repeated at 0.02x speed with the regrips annotated and the tracked locations of the left (red) and right (green) D3s indicated.
(D) A example of a sniffing maneuver with a wheat berry as viewed from below, shown at actual speed, then repeated at 0.1x speed with a pause at the point of minimum D3-nose distance and the tracked locations of the left D3 (red), right D3 (blue), and nose (green) indicated.
(MP4)

**S4 Video. Feeding mice cycle rapidly between oromanual and holding phases of food-handling.** Two examples of a mouse feeding on a flaxseed, showing alternation between oromanual and holding phases at actual speed, then repeated at 0.1x speed with annotations to indicate the oromanual and holding phases. In the second example, the tracked positions of the digits and nose are also shown.
(MP4)

**S5 Video. Rapid regrip maneuver during oromanual handling.** Three examples of brief periods of oromanual handling, each including one or more regrip maneuvers. Each example is shown at actual speed, then repeated at 0.02x speed with the regrips annotated and the tracked locations of the left (red) and right (green) D3s indicated.
(MP4)

**S6 Video. Rapid hold-to-nose "sniffing" maneuver.** Three examples of a sniffing maneuver as viewed from below. Each example is shown at actual speed, then repeated at 0.1x speed with a pause at the point of minimum D3-nose distance and the tracked locations of the left D3 (red), right D3 (blue), and nose (green) indicated.
(MP4)

**S7 Video. Complex movements during oromanual handling.** An example of oromanual handling including a full 180˚ rotation of the seeding, shown at 1x speed, then repeated at 0.1x speed with the rotation itself shown at 0.01x speed. Note the involvement of the thumb at various points, including being used as a fulcrum about which the seed is rotated, and to push one end of the seed around while the other is held in the mouth.
(MP4)

**S8 Video. Double-holding.** Example 1 shows a period where the seed (a wheat berry) is broken in two and both pieces simultaneously held, displayed first at 1x speed, then repeated at 0.1x speed. After the seed breaks into two fragments, one is held between the D1s in a thumbhold and the other is held against the palm by D3 and D4.
Example 2 shows another double-holding example, this time involving couscous, first at 1x speed, then repeated at 0.02x speed. Two unimanual regrips are highlighted, the first in which the larger fragment is held in the regripping hand, and the second in which it is held in the non-regripping hand.
Example 3 shows a third double-holding example, first at 1x speed, then repeated at 0.1x speed. Here, the larger fragment falls when the seed breaks, but the mouse is able to catch it. Also highlighted are two regrips involving primarily the D2s and the use of D4-5 to press the larger fragment towards the mouth after the first is consumed.
(MP4)

**S9 Video. Seed handling by head-fixed mice.** (A) An example of a head-fixed mouse holding a flaxseed in with various grip types shown at actual speed, then repeated at 0.1x speed with the different grip types annotated.
(B) An example of a mouse feeding on a flaxseed while head-fixed shown at actual speed, then repeated at 0.1x speed, with annotations to indicate the oromanual and holding phases.
(C) An example of a brief period of oromanual handling of a flaxseed by a head-fixed mouse including several regrip maneuvers shown at actual speed, then repeated at 0.02x speed with the regrips annotated and the tracked locations of the left (red) and right (green) D3s indicated.
(D) A example of a sniffing maneuver by a head-fixed mouse as viewed from below shown at actual speed, then repeated at 0.1x speed with a pause at the point of minimum D3-nose distance and the tracked locations of the left D3 (red), right D3 (blue), and nose (green) indicated.
(MP4)

**S10 Video. D1 use by squirrels during feeding.** Two examples of squirrels feeding on a peanut, shown at actual speed, then repeated at 0.1x speed with the oromanual handling and thumb-holding phases annotated. The thumb cleft and examples of a regrip and a sniff are also highlighted.
(MP4)

## Acknowledgments

We thank G. Galiñanes, D. Huber, L. Miller, A. Miri, M. Tresch, and I. Whishaw for discussions and advice, K. Guo and N. Yamawaki for many helpful comments and suggestions, L.

Lambot for assistance with imaging experiments, and D. Procissi of the Northwestern University Center for Translational Imaging for micro-CT imaging and rendering.

## Author Contributions

**Conceptualization:** Gordon M. G. Shepherd.

**Data curation:** John M. Barrett, Gordon M. G. Shepherd.

**Formal analysis:** John M. Barrett, Gordon M. G. Shepherd.

**Funding acquisition:** Gordon M. G. Shepherd.

**Investigation:** John M. Barrett, Martinna G. Raineri Tapies, Gordon M. G. Shepherd.

**Methodology:** John M. Barrett, Gordon M. G. Shepherd.

**Project administration:** Gordon M. G. Shepherd.

**Resources:** Gordon M. G. Shepherd.

**Software:** John M. Barrett.

**Supervision:** Gordon M. G. Shepherd.

**Visualization:** John M. Barrett, Gordon M. G. Shepherd.

**Writing – original draft:** John M. Barrett, Gordon M. G. Shepherd.

**Writing – review & editing:** Martinna G. Raineri Tapies.

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
