## [Decision Letter · Decision Letter 0]

30 Oct 2019

PONE-D-19-26076

Manual dexterity of mice during food-handling involves the thumb and a set of fast basic movements

PLOS ONE

Dear Dr Barrett,

Thank you for submitting your manuscript to PLOS ONE. After careful consideration, we feel that it has merit but does not fully meet PLOS ONE’s publication criteria as it currently stands. Therefore, we invite you to submit a revised version of the manuscript that addresses the points raised during the review process.

We would appreciate receiving your revised manuscript by Dec 14 2019 11:59PM. To enhance the reproducibility of your results, we recommend that if applicable you deposit your laboratory protocols in protocols.io, where a protocol can be assigned its own identifier (DOI) such that it can be cited independently in the future. For instructions see: http://journals.plos.org/plosone/s/submission-guidelines#loc-laboratory-protocols

We look forward to receiving your revised manuscript.

Kind regards,

Lesley Joy Rogers, B.Sc. (Hons), D.Phil., D.Sc.

Academic Editor

PLOS ONE

Journal Requirements:

"This work was funded by grant NS061963 from the National Institute of Neurological Disorders

636 and Stroke at the National Institutes of Health. The funders had no role in study design, data

637 collection and analysis, decision to publish, or preparation of the manuscript.".

 "The funders had no role in study design, data collection and analysis, decision to publish, or preparation of the manuscript.".

Additional Editor Comments (if provided):

Thank you for submitting your paper to PLoS One. It requires some revision. Please attend to all of the points raised by the reviewers when you revise the manuscript.

Reviewers' comments:

Reviewer's Responses to Questions

**Comments to the Author**

1. Is the manuscript technically sound, and do the data support the conclusions?

Reviewer #1: Yes

Reviewer #2: Partly

2. Has the statistical analysis been performed appropriately and rigorously? 

Reviewer #1: Yes

Reviewer #2: I Don't Know

3. Have the authors made all data underlying the findings in their manuscript fully available?

Reviewer #1: Yes

Reviewer #2: No

4. Is the manuscript presented in an intelligible fashion and written in standard English?

Reviewer #1: Yes

Reviewer #2: Yes

5. Review Comments to the Author

Reviewer #1: The paper conducts a detailed analysis of the uses of the thumb by mice during feeding on foods that can be held in the paws. The analyses reveal that despite their apparent vestigial structure, the thumbs have specific modes of holding and are highly dexterous. Most critical, the movements used can be deconstructed into discrete, kinematic elements that remain robust even when the mouse’s head is constrained from moving. The details on how kinematic elements transition across phases of the feeding cycle provides a methodological advance in the field and will enable for more nuanced cross-species comparisons and for the analysis of the effects of neurological damage. I have no substantive concerns with the methods of data collection and analyses. I have just a couple of comments concerning clarity and updating the references.

Line 267: As the Introduction was focused on making a case for why it is of value to gain greater descriptive understanding of the use of D1 in mice, this section on squirrels appears unexpectedly. That the study includes comparison with squirrels should be mentioned, with the appropriate rationale, should be noted in the Introduction. Moreover, while in the Methods, the approval for use of the squirrels is mentioned, no information is provided as to the sex and age of the two animals analyzed. If the sex is unknown, it should be so stated, and at least an estimate as to whether these animals were adults or immature should be provided.

Lines 268-269: This statement is no longer correct, as there is another, more recently published study reporting bimanual thumb holding in species of lemurs (Pellis, S. M., & Pellis, V. C. (2012). Anatomy is important, but need not be destiny: Novel uses of the thumb in aye-ayes compared to other lemurs. Behavioural Brain Research, 231, 378-285). There are still good reasons, such as phylogenetic closeness, to restrict the comparison between mice and squirrels for this paper, but to embed the current study within the current literature, the referencing should be updated accordingly. This is particularly important for how the origins of bimanual thumb holding, is interpreted (see lines 334-335). The occurrence of such a hold in some rodents and some primates would suggest that this hold has been independently evolved multiple times.

Reviewer #2: The manuscript by John Martin Barrett et al. thoroughly analyzes the fine movements of forelimbs involved in food-handling, with a particular focus on the role of D1. They carefully dissect the handling grips and the motion sequences related to food holding, providing a valuable contribution to the analysis of fine movements, whose application could prove extremely useful in animal models of pathologies characterized by impaired dexterous movements.

Major concerns

1. The study would benefit from a broader introduction. The methods here described could in fact be applied to mouse models of disorders characterized by deficient purposeful hand movements (see e.g. Neural Plast. 2015;2015:326184. doi: 10.1155/2015/326184. Epub 2015 Jun 22. Deficient Purposeful Use of Forepaws in Female Mice Modelling Rett Syndrome). I thus suggest to introduce the translational (in addition to the evolutionary) relevance of studying the dexterous movement of forelimbs in rodents.

2. Please provide statistical outputs throughout the results section to help the reader appreciate the magnitude of the described effects. This is particularly relevant given that the number of animals used is very limited (and not justified by an a priori evaluation of sample size) and the experimental animals are not homogeneous at all: there is a disproportionate use of male and female, mice are between 2 and 6 months and their weight is very variable.

3. The decision to analyse male and females without considering the effects of sex should be truly justified in the text.

4. Please check in the methods section the weights of the animals (weighing 14.5-28.1 g at the time

of video recordings). I suggest that the lower body weight declared may be a misprint. In my experience mice weighting 14 grams at adulthood are not healthy and, of course, I would not expose mice with such a low body weight to food deprivation.

5. Please provide a justification for carry out the analysis on head fixed mice. Given the severity of the procedure, a deeper justification should be provided.

6. Please correct this sentence: “Squirrels are the only mammals previously noted to use thumb-holding to handle food [11]” I immagine you meant not primate mammals. Please provide a clearer justification for the need of comparison between mice and squirrels.

Minor points:

Table 1

- Line 94: please specify what “freely moving” means (here or in line 72).

- Line 95: please add (#videos) after “the number of video segment used for the analysis is listed”.

- Line 95: is the average or the median length of the video specified in the table?

- Please specify M.A.D.

- Is the total time and total average of videos analyzed independently from each animal relevant? (last three rows of the column)

- Did the authors consider the total length/number of video analyzed for each animal in the correlations analyses?

Figure 2

- The difference between E and C seems to be mainly given by the pellet dimension

- Lines 106-107: please add the specifications for grey lines and purple line as for figure 2D

Figure 3

- Line 131: Please add the specification “median ± mad”

- Line 132: please substitute “error bars” with “thick line” to be consistent with figure 2

- Line 137: does “the example” refer to a single animal? Please specify (see lines 185-186)

- Line 145: please substitute “error bars” with “thick line” to be consistent with figure 2

Figure 4

- Line 193: please substitute “error bars” with “thick line”

Figure 5

- Line 213: plot of L or of L ventral? Please specify

Figure 7:

- Line 313: Please, explain the statistical reason for treating the repeated observations as independent samples and for pooling together the results from the two squirrels. The number of video segments analyzed for each individual are indeed a within-subject factor, to be distinguished from the between-subject factor represented by the individuals’ number.

- Line 317-318: please specify if section O refers to one single squirrel or two, and which signales are averaged.

Line 157: please add “(L)” after “D3-to-nose dimension”

Line 160: please add berries to the end of the sentence

Line 175: It would be interesting to analyze unimanual grips in relation to the animals’ handedness. Could you comment on that?

Page 12, section: complex movements

Are there objective parameters useful to evaluate the complex movements observed? Please discuss.

Line 269: Please rephrase.

Lines 273-275: wouldn’t the anatomical description be part of the reason why you looked at the squirrels (anatomy similar to mice)? If so, I would suggest to move this sentence to the beginning of the paragraph.

Page 17 line 341: please, comment also on the possible influence of handedness on symmetry/asymmetry of forelimb movements

Overall comment to results:

Please provide statistical information throughout the results section

Discussion

- Line 366: please provide examples.

Materials and methods

- Line 378: please add a comment of the use of both sexes in different proportion for each experiment.

- Line 410: please specify the lux.

- Lines 438-443: I would suggest to move them to the supplementary section.

- Lines 455-456: this information is not needed in this section.

- Ethograms section: were the video manually evaluated by only one ore multiple observers? Please specify.

6. PLOS authors have the option to publish the peer review history of their article (what does this mean?). If published, this will include your full peer review and any attached files.

Reviewer #1: No

Reviewer #2: No

---

## [Author Response · Author response to Decision Letter 0]

12 Nov 2019

Response to Reviewers

We thank the reviewers for their careful reading of the manuscript and their constructive criticism. Below we address the concerns and suggestions in a point-by-point manner. In the red-lined version of the revised manuscript we have tracked all the substantive changes, but not minor/trivial ones relating to journal style or formatting.

Reviewer #1: The paper conducts a detailed analysis of the uses of the thumb by mice during feeding on foods that can be held in the paws. The analyses reveal that despite their apparent vestigial structure, the thumbs have specific modes of holding and are highly dexterous. Most critical, the movements used can be deconstructed into discrete, kinematic elements that remain robust even when the mouse’s head is constrained from moving. The details on how kinematic elements transition across phases of the feeding cycle provides a methodological advance in the field and will enable for more nuanced cross-species comparisons and for the analysis of the effects of neurological damage. I have no substantive concerns with the methods of data collection and analyses. I have just a couple of comments concerning clarity and updating the references.

Line 267: As the Introduction was focused on making a case for why it is of value to gain greater descriptive understanding of the use of D1 in mice, this section on squirrels appears unexpectedly. That the study includes comparison with squirrels should be mentioned, with the appropriate rationale, should be noted in the Introduction. Moreover, while in the Methods, the approval for use of the squirrels is mentioned, no information is provided as to the sex and age of the two animals analyzed. If the sex is unknown, it should be so stated, and at least an estimate as to whether these animals were adults or immature should be provided.

RESPONSE: Thank you for pointing this out. Since the study is almost entirely about mice, we had thought that the sentence in the Abstract about the squirrel studies would suffice as a kind of introduction, but we can see how mentioning this in the Introduction will be beneficial. The relevant sentences are now in their own final paragraph of the Intro, which includes additional points to address concerns raised by Reviewer #2. In the field studies of squirrels, as the reviewer correctly points out we were unable to definitively determine the sex and age of the animals. Following the reviewer’s suggestion this is now mentioned in the Methods/Animals section.

Lines 268-269: This statement is no longer correct, as there is another, more recently published study reporting bimanual thumb holding in species of lemurs (Pellis, S. M., & Pellis, V. C. (2012). Anatomy is important, but need not be destiny: Novel uses of the thumb in aye-ayes compared to other lemurs. Behavioural Brain Research, 231, 378-285). There are still good reasons, such as phylogenetic closeness, to restrict the comparison between mice and squirrels for this paper, but to embed the current study within the current literature, the referencing should be updated accordingly. This is particularly important for how the origins of bimanual thumb holding, is interpreted (see lines 334-335). The occurrence of such a hold in some rodents and some primates would suggest that this hold has been independently evolved multiple times.

RESPONSE: We thank the reviewer for this point. Reviewer #2 also made a similar point about this sentence. We have changed it to read: “… squirrels are the only rodents previously noted to use bimanual thumb-holding …”. We have also clarified the related section in the second paragraph of the Discussion, and now mention and cite the lemur paper. We also mention the aye-aye’s very recently described ‘pseudo-thumbs’, which intriguingly may be involved in thumb-holding in a manner that conceivably parallels the way the thenar pads of mice and squirrels are involved in bimanual gripping. 

Reviewer #2: The manuscript by John Martin Barrett et al. thoroughly analyzes the fine movements of forelimbs involved in food-handling, with a particular focus on the role of D1. They carefully dissect the handling grips and the motion sequences related to food holding, providing a valuable contribution to the analysis of fine movements, whose application could prove extremely useful in animal models of pathologies characterized by impaired dexterous movements.

Major concerns

1. The study would benefit from a broader introduction. The methods here described could in fact be applied to mouse models of disorders characterized by deficient purposeful hand movements (see e.g. Neural Plast. 2015;2015:326184. doi: 10.1155/2015/326184. Epub 2015 Jun 22. Deficient Purposeful Use of Forepaws in Female Mice Modelling Rett Syndrome). I thus suggest to introduce the translational (in addition to the evolutionary) relevance of studying the dexterous movement of forelimbs in rodents.

RESPONSE: Yes, and we indeed already explicitly mentioned the translational potential in the final paragraph of the Discussion, and cited this paper among others. We feel that the Discussion, rather than the Introduction, is the appropriate place to discuss this aspect of our study. Furthermore, we add emphasis to this aspect by placing it in its own paragraph at the very end of the Discussion. We do agree that the Introduction could be expanded somewhat, and have separated the final part into a new last paragraph. In the Intro, after the sentence “Mice are widely used to study forelimb function and dysfunction”, we also now cite the review article by Klein et al. (2012, “The use of rodent skilled reaching as a translational model for investigating brain damage and disease”).

2. Please provide statistical outputs throughout the results section to help the reader appreciate the magnitude of the described effects. This is particularly relevant given that the number of animals used is very limited (and not justified by an a priori evaluation of sample size) and the experimental animals are not homogeneous at all: there is a disproportionate use of male and female, mice are between 2 and 6 months and their weight is very variable.

RESPONSE: Statistical details are now provided throughout. Points regarding sex and weight are addressed below.

3. The decision to analyse male and females without considering the effects of sex should be truly justified in the text.

RESPONSE: As sex-dependent differences were not anticipated for food-handling behavior, the study was not specifically designed to investigate such differences. However, following current NIH policy, we included both males and females in the study. (The use of animals of only one sex, which the reviewer seems to be suggesting, is now strongly discouraged by NIH: “Strong justification from the scientific literature, preliminary data, or other relevant considerations must be provided for applications proposing to study only one sex”; see e.g. https://orwh.od.nih.gov/sex-gender/nih-policy-sex-biological-variable.) The somewhat imbalanced sex ratios across cohorts reflects the particular availability of mice in our colony at the time of experiments. We saw no trends or indications of sex-dependent differences in food-handling behavior, but had we done so we would have (again consistent with NIH policy) expanded the study to investigate this in detail. The first paragraph of the Methods has been modified to clarify these sex-related issues.

4. Please check in the methods section the weights of the animals (weighing 14.5-28.1 g at the time of video recordings). I suggest that the lower body weight declared may be a misprint. In my experience mice weighting 14 grams at adulthood are not healthy and, of course, I would not expose mice with such a low body weight to food deprivation.

RESPONSE: Thank you for pointing this out. The weights listed are post food-restriction. This has been clarified and pre-restriction body weights added. Moreover, the mouse in question was a female at the younger end of our cohort (~9 weeks at the time of recordings) with a pre-restriction bodyweight of 17.4 g, within the normal range for C57BL/6 mice (https://www.jax.org/jax-mice-and-services/strain-data-sheet-pages/body-weight-chart-000664). Finally, all mice were monitored throughout the study for signs of ill health (including body condition scoring) and none were observed.

5. Please provide a justification for carry out the analysis on head fixed mice. Given the severity of the procedure, a deeper justification should be provided.

RESPONSE: We now include this justification in the last paragraph of the Introduction.

6. Please correct this sentence: “Squirrels are the only mammals previously noted to use thumb-holding to handle food [11]” I immagine you meant not primate mammals. Please provide a clearer justification for the need of comparison between mice and squirrels.

RESPONSE: Please see our response to Reviewer #1’s second comment.

Minor points:

Table 1

- Line 94: please specify what “freely moving” means (here or in line 72).

RESPONSE: Freely moving as opposed to head-fixed. We have changed this to “main cohort” in the text and clarified which figures and videos this refers to.

- Line 95: please add (#videos) after “the number of video segment used for the analysis is listed”.

RESPONSE: We have updated the table headings to match the legend.

- Line 95: is the average or the median length of the video specified in the table?

RESPONSE: Median. This has been clarified.

- Please specify M.A.D.

RESPONSE: Median absolute deviation. We have defined this at the first occurrence in the text.

- Is the total time and total average of videos analyzed independently from each animal relevant? (last three rows of the column)

RESPONSE: We have deleted the last two rows.

- Did the authors consider the total length/number of video analyzed for each animal in the correlations analyses?

RESPONSE: The only correlation analysis involving multiple mice reported in the paper is that comparing grip-type to seed size, for which an identical number of holding periods was considered for each animal and seed type. This has been clarified in the text at the end of the first section of the Results.

Figure 2

- The difference between E and C seems to be mainly given by the pellet dimension

RESPONSE: Thank you for this observation. We feel that the analysis shown in panel F addresses the effect of pellet dimension on grip type.

- Lines 106-107: please add the specifications for grey lines and purple line as for figure 2D

RESPONSE: Done

Figure 3

- Line 131: Please add the specification “median ± mad”

RESPONSE: Done, by mentioning this once after the descriptions of the two plots, as it applies to both.

- Line 132: please substitute “error bars” with “thick line” to be consistent with figure 2

RESPONSE: Done, but by mentioning the gray/purple colors, not thicknesses.

- Line 137: does “the example” refer to a single animal? Please specify (see lines 185-186)

RESPONSE: Yes. This has been clarified.

- Line 145: please substitute “error bars” with “thick line” to be consistent with figure 2

RESPONSE: Done

Figure 4

- Line 193: please substitute “error bars” with “thick line”

RESPONSE: Done

Figure 5

- Line 213: plot of L or of L ventral? Please specify

RESPONSE: Lventral, this has been clarified.

Figure 7:

- Line 313: Please, explain the statistical reason for treating the repeated observations as independent samples and for pooling together the results from the two squirrels. The number of video segments analyzed for each individual are indeed a within-subject factor, to be distinguished from the between-subject factor represented by the individuals’ number.

RESPONSE: The reviewer makes a valid point. We have replaced the descriptive statistics in the text with medians first within and then between animals. For the figure, we have kept all data points but color-coded them by animal and additionally plotted the within-animal medians. 

- Line 317-318: please specify if section O refers to one single squirrel or two, and which signales are averaged.

RESPONSE: Single squirrel. This has been clarified.

Line 157: please add “(L)” after “D3-to-nose dimension”

RESPONSE: Done. 

Line 160: please add berries to the end of the sentence

RESPONSE: Done.

Line 175: It would be interesting to analyze unimanual grips in relation to the animals’ handedness. Could you comment on that?

RESPONSE: We have added a comment to this effect in the Discussion. 

Page 12, section: complex movements

Are there objective parameters useful to evaluate the complex movements observed? Please discuss.

RESPONSE: We explored this but did not find dramatic differences in e.g. L or D during complex movements compared to the oromanual phase in general. Nevertheless, we highlight this as an area for future study in the penultimate paragraph of the discussion. 

Line 269: Please rephrase.

RESPONSE: Done.

Lines 273-275: wouldn’t the anatomical description be part of the reason why you looked at the squirrels (anatomy similar to mice)? If so, I would suggest to move this sentence to the beginning of the paragraph.

RESPONSE: Not exactly; the motivation was the observations of squirrel feeding behavior in the Whishaw reference, as mentioned in the first sentence of the paragraph. 

Page 17 line 341: please, comment also on the possible influence of handedness on symmetry/asymmetry of forelimb movements

RESPONSE: We did not determine the handedness, so we mention this as an area for future studies. 

Overall comment to results:

Please provide statistical information throughout the results section

RESPONSE: See earlier response to same point. 

Discussion

- Line 366: please provide examples.

RESPONSE: Done. 

Materials and methods

- Line 378: please add a comment of the use of both sexes in different proportion for each experiment.

RESPONSE: Addressed above.

- Line 410: please specify the lux.

RESPONSE: We are unable to do so, as the manufacturer does not provide this information and we did not measure the light intensity at the level of the mouse’s hands during feeding. However, we have added the part number for the LEDs to the Methods.

- Lines 438-443: I would suggest to move them to the supplementary section.

RESPONSE: We appreciate the suggestion, but would prefer to keep this brief description where it is in Methods, both because (a) it is relevant for this head-fixation section, and (b) we do not have a separate supplementary section, only supplementary figures; i.e., moving it would entail adding an entire stand-alone supplemental methods section for just this one point. The associated figure, Fig. S3, is already in supplemental format.

- Lines 455-456: this information is not needed in this section.

RESPONSE: We have removed it, and added a couple of details from it to the legend of Table 1.

- Ethograms section: were the video manually evaluated by only one ore multiple observers? Please specify.

RESPONSE: Thank you for pointing this out. We have clarified this in the “Methods/Ethograms” section.

-----------------

Other changes:

Discussion, 4th paragraph – As this paragraph regarding oromanual/holding phases was rather brief, we have added two more sentences on this aspect.

---

## [Decision Letter · Decision Letter 1]

28 Nov 2019

PONE-D-19-26076R1

Manual dexterity of mice during food-handling involves the thumb and a set of fast basic movements

PLOS ONE

Dear Dr Barrett,

Thank you for submitting your manuscript to PLOS ONE. After careful consideration, we feel that it has merit but does not fully meet PLOS ONE’s publication criteria as it currently stands. Therefore, we invite you to submit a revised version of the manuscript that addresses the points raised during the review process.

We would appreciate receiving your revised manuscript by Jan 12 2020 11:59PM. To enhance the reproducibility of your results, we recommend that if applicable you deposit your laboratory protocols in protocols.io, where a protocol can be assigned its own identifier (DOI) such that it can be cited independently in the future. For instructions see: http://journals.plos.org/plosone/s/submission-guidelines#loc-laboratory-protocols

We look forward to receiving your revised manuscript.

Kind regards,

Lesley Joy Rogers, B.Sc. (Hons), D.Phil., D.Sc.

Academic Editor

PLOS ONE

Additional Editor Comments (if provided):

You have improved your paper by making revisions, but Reviewer 2 requests that you make further revisions and I agree. Please address all of the points made by Reviewer 2.

Reviewers' comments:

Reviewer's Responses to Questions

**Comments to the Author**

1. If the authors have adequately addressed your comments raised in a previous round of review and you feel that this manuscript is now acceptable for publication, you may indicate that here to bypass the “Comments to the Author” section, enter your conflict of interest statement in the “Confidential to Editor” section, and submit your "Accept" recommendation.

Reviewer #1: All comments have been addressed

Reviewer #2: (No Response)

2. Is the manuscript technically sound, and do the data support the conclusions?

Reviewer #1: Yes

Reviewer #2: Yes

3. Has the statistical analysis been performed appropriately and rigorously? 

Reviewer #1: Yes

Reviewer #2: Yes

4. Have the authors made all data underlying the findings in their manuscript fully available?

Reviewer #1: Yes

Reviewer #2: Yes

5. Is the manuscript presented in an intelligible fashion and written in standard English?

Reviewer #1: Yes

Reviewer #2: Yes

6. Review Comments to the Author

Reviewer #1: The revised paper satisfactorily addresses all the concerns raised regarding the original version. I have no further concerns. This contribution sets a new standard for studying the microstructure of paw use in food handling and so will be an important guide for future studies.

Reviewer #2: I do still have some concerns, since few of the issues that I raised were not fully addressed.

Please see below my further comments.

1. The study would benefit from a broader introduction. The methods here described could in

fact be applied to mouse models of disorders characterized by deficient purposeful hand

movements (see e.g. Neural Plast. 2015;2015:326184. doi: 10.1155/2015/326184. Epub 2015 Jun

22. Deficient Purposeful Use of Forepaws in Female Mice Modelling Rett Syndrome). I thus

suggest to introduce the translational (in addition to the evolutionary) relevance of studying the

dexterous movement of forelimbs in rodents.

YOUR RESPONSE: Yes, and we indeed already explicitly mentioned the translational potential in

the final paragraph of the Discussion, and cited this paper among others. We feel that the

Discussion, rather than the Introduction, is the appropriate place to discuss this aspect of

our study. Furthermore, we add emphasis to this aspect by placing it in its own paragraph

at the very end of the Discussion. We do agree that the Introduction could be expanded

somewhat, and have separated the final part into a new last paragraph. In the Intro, after

the sentence “Mice are widely used to study forelimb function and dysfunction”, we also now cite the review article by Klein et al. (2012, “The use of rodent skilled reaching as a

translational model for investigating brain damage and disease”).

MY FURTHER COMMENTS: I still have concerns about the introduction. In my opinion this section would benefit from a clear contextualization of the aims of the present work in both the evolutionary and translational fields. To what extent these field would benefit from a broader understanding of mice fine forehand movements?

3. The decision to analyse male and females without considering the effects of sex should be

truly justified in the text.

YOUR RESPONSE: As sex-dependent differences were not anticipated for food-handling

behavior, the study was not specifically designed to investigate such differences. However,

following current NIH policy, we included both males and females in the study. (The use of

animals of only one sex, which the reviewer seems to be suggesting, is now strongly

discouraged by NIH: “Strong justification from the scientific literature, preliminary data,

or other relevant considerations must be provided for applications proposing to study only

one sex”; see e.g. https://orwh.od.nih.gov/sex-gender/nih-policy-sex-biological-variable.)

The somewhat imbalanced sex ratios across cohorts reflects the particular availability of

mice in our colony at the time of experiments. We saw no trends or indications of sex-dependent differences in food-handling behavior, but had we done so we would have (again

consistent with NIH policy) expanded the study to investigate this in detail. The first

paragraph of the Methods has been modified to clarify these sex-related issues.

[Sex-dependent differences were not anticipated; the study was not designed to test for these and was under-powered to do so. Because de-aggregation of the data by sex did not reveal any obvious sex-dependent differences, the data were pooled. ]

MY FURTHER COMMENTS: Please, clarify what you mean with “anticipated” in the text: is there no evidence in the literature? If de-aggregation of the data by sex was done to unravel possible sex-dependent differences, why you declare that the study was underpowered to do so?

4. Please check in the methods section the weights of the animals (weighing 14.5-28.1 g at the

time of video recordings). I suggest that the lower body weight declared may be a misprint. In

my experience mice weighting 14 grams at adulthood are not healthy and, of course, I would not

expose mice with such a low body weight to food deprivation.

YOUR RESPONSE: Thank you for pointing this out. The weights listed are post food-restriction.

This has been clarified and pre-restriction body weights added. Moreover, the mouse in

question was a female at the younger end of our cohort (~9 weeks at the time of recordings)

with a pre-restriction bodyweight of 17.4 g, within the normal range for C57BL/6 mice

(https://www.jax.org/jax-mice-and-services/strain-data-sheet-pages/body-weight-chart-

000664). Finally, all mice were monitored throughout the study for signs of ill health

(including body condition scoring) and none were observed.

MY FURTHER COMMENTS: Including the information about the animal’s health monitoring in the methods section is necessary, given the food deprivation procedure. Please, also provide evidence for age/weight not influencing the parameters measured.

5. Lines 408-424. Please rephrase: it is indeed interesting to address the open points of such works, but such a list risks to reduce the relevance of each point outlined.

7. PLOS authors have the option to publish the peer review history of their article (what does this mean?). If published, this will include your full peer review and any attached files.

Reviewer #1: No

Reviewer #2: No

---

## [Author Response · Author response to Decision Letter 1]

1 Dec 2019

Response to Reviewer 2

“… I still have concerns about the introduction. In my opinion this section would benefit from a clear contextualization of the aims of the present work in both the evolutionary and translational fields. To what extent these field would benefit from a broader understanding of mice fine forehand movements?”

RESPONSE: We have added a sentence to the Introduction discussing this. 

“Please, clarify what you mean with “anticipated” in the text: is there no evidence in the literature? If de-aggregation of the data by sex was done to unravel possible sex-dependent differences, why you declare that the study was underpowered to do so?”

RESPONSE: Yes, that is the intended meaning of “anticipated”. Underpowered simply means it is impossible to reach statistical significance, but it is still possible to visually inspect the data for trends related to sex. We have edited the paragraph to provide a clearer explanation of the point about sex differences. 

“Including the information about the animal’s health monitoring in the methods section is necessary, given the food deprivation procedure. Please, also provide evidence for age/weight not influencing the parameters measured.”

RESPONSE: We have included this information in the Methods section. We also include statements about the lack of age- or weight-dependent correlations in the data. 

“Lines 408-424. Please rephrase: it is indeed interesting to address the open points of such works, but such a list risks to reduce the relevance of each point outlined.”

RESPONSE: We have considered ways to rephrase this section, but have decided that we prefer to keep this part of the Discussion as it is; it concisely conveys what we consider to be some of the most interesting questions for future studies to address.

---

## [Editor Report · Decision Letter 2]

6 Dec 2019

Manual dexterity of mice during food-handling involves the thumb and a set of fast basic movements

PONE-D-19-26076R2

Dear Dr. Barrett,

We are pleased to inform you that your manuscript has been judged scientifically suitable for publication and will be formally accepted for publication once it complies with all outstanding technical requirements.

With kind regards,

Lesley Joy Rogers, B.Sc. (Hons), D.Phil., D.Sc.

Academic Editor

PLOS ONE

Additional Editor Comments (optional):

The authors have made all of the important revisions to their paper.
---

## [Editor Report · Acceptance letter]

10 Dec 2019

PONE-D-19-26076R2 

Manual dexterity of mice during food-handling involves the thumb and a set of fast basic movements 

Dear Dr. Barrett:

I am pleased to inform you that your manuscript has been deemed suitable for publication in PLOS ONE. Congratulations! Your manuscript is now with our production department. 

With kind regards,

on behalf of

Prof. Lesley Joy Rogers 

Academic Editor

PLOS ONE